# Competition between electron pairing and phase coherence in superconducting interfaces

G. Singh[1,2], A. Jouan[1,2], L. Benfatto [3,4], F. Couëdo[1,2], P. Kumar[5], A. Dogra[5], R.C. Budhani[6], S. Caprara [3,4] M. Grilli [3,4], E. Lesne[7], A. Barthélémy[7], M. Bibes [7], C. Feuillet-Palma[1,2], J. Lesueur [1,2] & N. Bergeal [1,2]

In LaAlO$_3$/SrTiO$_3$ heterostructures, a gate tunable superconducting electron gas is confined in a quantum well at the interface between two insulating oxides. Remarkably, the gas coexists with both magnetism and strong Rashba spin–orbit coupling. However, both the origin of superconductivity and the nature of the transition to the normal state over the whole doping range remain elusive. Here we use resonant microwave transport to extract the superfluid stiffness and the superconducting gap energy of the LaAlO$_3$/SrTiO$_3$ interface as a function of carrier density. We show that the superconducting phase diagram of this system is controlled by the competition between electron pairing and phase coherence. The analysis of the superfluid density reveals that only a very small fraction of the electrons condenses into the superconducting state. We propose that this corresponds to the weak filling of high-energy $d_{xz}/d_{yz}$ bands in the quantum well, more apt to host superconductivity.

[1] Laboratoire de Physique et d'Etude des Matériaux, ESPCI Paris, PSL Research University, CNRS, 10 Rue Vauquelin, 75005 Paris, France. [2] Université Pierre and Marie Curie, Sorbonne-Universités, 75005 Paris, France. [3] Institute for Complex Systems (ISC-CNR), UOS Sapienza, Piazzale A. Moro 5, 00185 Roma, Italy. [4] Dipartimento di Fisica Università di Roma "La Sapienza", Piazzale A. Moro 5, 00185 Roma, Italy. [5] National Physical Laboratory, Council of Scientific and Industrial Research (CSIR), Dr. K.S. Krishnan Marg, New Delhi, 110012, India. [6] Condensed Matter Low Dimensional Systems Laboratory, Department of Physics, Indian Institute of Technology, Kanpur, 208016, India. [7] Unité Mixte de Physique CNRS-Thales, 1 Av. A. Fresnel, 91767 Palaiseau, France. G. Singh and A. Jouan contributed equally to this work. Correspondence and requests for materials should be addressed to L.B. (email: lara.benfatto@roma1.infn.it) or to N.B. (email: nicolas.bergeal@espci.fr)

The superconducting phase diagram of $LaAlO_3/SrTiO_3$ interfaces defined by plotting the critical temperature $T_c$ as a function of electrostatic doping has the shape of a dome. It ends into a quantum critical point, where the $T_c$ is reduced to zero, as carriers are removed from the interfacial quantum well[1–4]. Despite a few proposals[5–7], the origin of this carrier density dependence, and in particular the non-monotonic suppression of $T_c$, remains unclear. To investigate this issue, one must consider the two fundamental energy scales associated with superconductivity. On the one hand, the gap energy $\Delta$ measures the pairing strength between electrons that form Cooper pairs. On the other hand, the superfluid stiffness $J_s$ determines the cost of a phase twist in the superconducting condensate. In conventional superconductors, well described by Bardeen–Cooper–Schrieffer (BCS) theory, $J_s$ is much higher than $\Delta$ and the superconducting transition is controlled by the breaking of Cooper pairs. However, when the stiffness is strongly reduced, phase fluctuations play a major role and the suppression of $T_c$ can be dominated by the loss of phase coherence[8]. Tunneling experiments in the low doping regime of $LaAlO_3/SrTiO_3$ interfaces evidenced the presence of a pseudogap in the density of states above $T_c$[9]. This can be interpreted as the signature of pairing surviving above $T_c$ while superconducting coherence is destroyed by strong phase fluctuations, enhanced by a low superfluid stiffness[10]. Superconductor-to-insulator quantum phase transitions driven by gate voltage[1] or magnetic field[11] also highlighted the predominant role of phase fluctuations in the suppression of $T_c$.

The two-dimensional (2D) superfluid density derived from the stiffness $\left(n_s^{2D} = \frac{4m}{\hbar^2} J_s\right)$ has to be analyzed within the context of the peculiar $LaAlO_3/SrTiO_3$ band structure. Under strong quantum confinement, the degeneracy of the $t_{2g}$ bands of $SrTiO_3$ ($d_{xy}$, $d_{xz}$, and $d_{yz}$ orbitals) is lifted, generating a rich and complex band structure[12,13]. The emergence of superconductivity for a given carrier density suggests that it could be intrinsically related to orbital occupancy in the interfacial quantum well. Experiments performed on (110)-oriented $LaAlO_3/SrTiO_3$ interfaces, for which the ordering of the $t_{2g}$ bands is reversed from that of the conventional (001) orientation, revealed that superconductivity behaves differently[14]. Instead of following the usual dome shape and disappearing at low doping, $T_c$ is only weakly affected by gating over a wide range of carrier density. This shows the important role of orbitals ordering and also suggests that only some specific bands could host superconductivity[15]. In particular, it has been emphasized that the $d_{xz}/d_{yz}$ band lying at high energy in the quantum well could play an important role because of its large density of states[6,7].

Here we use resonant microwave transport to measure the complex conductivity of the superconducting (001)-oriented $LaAlO_3/SrTiO_3$ interfaces. This allows us to directly extract the evolution of the superfluid stiffness in the phase diagram that we also convert into a gap energy through BCS theory in the dirty limit. Both energy scales are compared with theoretical predictions. The superfluid density $n_s$ deduced from $J_s$ is found to be close to the carrier density of the $d_{xz}/d_{yz}$ band extracted from multiband Hall effect measurements, highlighting the key role of this band in the emergence of superconductivity.

## Results

**Resonant microwave transport experiment**. In superconducting thin films, $J_s$ is usually assessed either from penetration depth measurements[16,17] or from dynamic transport measurements[18,19]. This latter method was adapted in this work for the specific case of $LaAlO_3/SrTiO_3$ samples. While superconductors have an infinite dc conductivity, they exhibit a finite complex

conductivity $\sigma(\omega)$ at non-zero frequency, which in 2D translates into a sheet conductance $G(\omega) = G_1(\omega) - iG_2(\omega)$. The real part $G_1(\omega)$ accounts for the transport of unpaired electrons existing at $T \neq 0$ and $\omega \neq 0$, and the imaginary part $G_2(\omega)$ accounts for the transport of Cooper pairs[20,21]. In the low-frequency limit $\hbar\omega \ll \Delta$, a superconductor behaves essentially as an inductor and $G_2(\omega) = \frac{1}{L_k\omega}$, where $L_k$ is the kinetic inductance of the superconductor due to the inertia of Cooper pairs[22]. The superfluid stiffness is directly related to the inductive response of the condensate through the relation $J_s = \frac{\hbar^2}{4e^2 L_k}$.

In this study, 8-uc-thick $LaAlO_3$ epitaxial layers were grown on $3 \times 3$ mm$^2$ $TiO_2$-terminated (001) $SrTiO_3$ single crystals by pulsed laser deposition (see Methods section). After the growth, a weakly conducting metallic back-gate of resistance $\sim$100 k$\Omega$ is deposited on the backside of the 200-μm- thick substrate. Figure 1 gives a schematic description of our experimental setup, inspired by recent developments in the field of quantum circuits[23,24]. The $LaAlO_3/SrTiO_3$ heterostructure is inserted in a microwave circuit board, between the central strip of a coplanar waveguide guide (CPW) transmission line and its ground. It is embedded into an RLC resonant circuit whose inductor $L_1$ and resistor $R_1$ are surface mounted microwave devices (SMDs), and whose capacitor $C_{STO}$ is due to the substrate in parallel with the 2D electron gas (2-DEG) (Fig. 1a, c). Because of the high dielectric constant of $SrTiO_3$ at low temperature (i.e., $\epsilon_r \simeq 24,000$), $C_{STO}$ dominates the circuit capacitance. More information on the sample environment can be found in the Supplementary Note 1 and Supplementary Fig. 1. A directional coupler is used to guide the microwave signal from port 1 to the sample through a bias-tee, and to separate the reflected signal which is amplified by a low-noise cryogenic high electron mobility transistor amplifier before reaching port 2 (Fig. 1b). The complex transmission coefficient $S_{21}(\omega)$ between the two ports is measured with a vector network analyzer. Standard microwave network analysis relates the reflection coefficient of the sample circuit $\Gamma(\omega)$ to $S_{21}(\omega)$ through complex error coefficients, which are determined by a calibration procedure (see Methods section and Supplementary Fig. 5). For a transmission line terminated by a circuit load of impedance $Z_L(\omega)$[25]

$$\Gamma(\omega) = \frac{A^{out}(\omega)}{A^{in}(\omega)} = \frac{Z_L(\omega) - Z_0}{Z_L(\omega) + Z_0}, \quad (1)$$

where $A^{in}$ and $A^{out}$ are the complex amplitudes of incident and reflected waves, and $Z_0 = 50\,\Omega$ is the characteristic impedance of the CPW transmission line. A reflection measurement gives therefore a direct access to the load impedance $Z_L(\omega)$ or equivalently its admittance $G_L(\omega) = 1/Z_L(\omega)$, commonly called complex conductance. In the present case, $Z_L(\omega)$ is the impedance of the RLC circuit represented in Fig. 1c, whose resonance frequency $\omega_0$ in the superconducting state is directly related to the kinetic inductance of the 2-DEG. Measuring $\omega_0$ as a function of gate voltage provides therefore a very direct method to determine the superfluid stiffness in the phase diagram. In addition, the setup of Fig. 1, which includes a bias-tee and protective capacitors in series with $L_1$ and $R_1$, allows measuring both the dc and ac microwave transport properties of the 2-DEG at the same time.

**Resonance in the normal and superconducting states**. After cooling the sample to 450 mK, the back-gate voltage is first swept to its maximum value +50 V while keeping the 2-DEG at the electrical ground, to ensure that no hysteresis will take place upon further gating[26]. In the limit $\omega \ll \tau^{-1}$ ($\tau$ is the elastic scattering time) and for temperatures higher than $T_c$, the 2-DEG behaves as

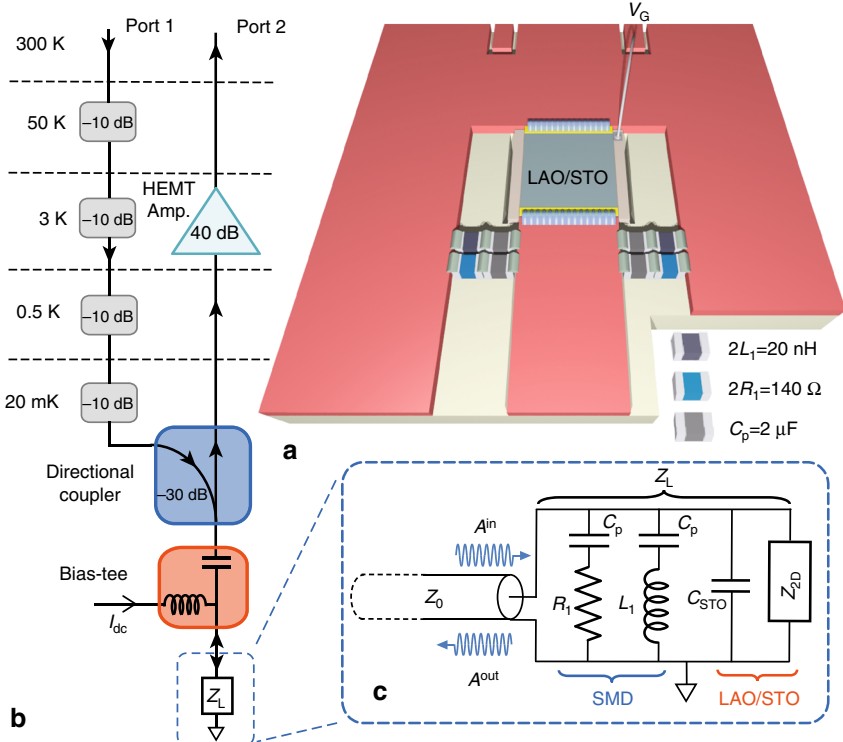

**Fig. 1** The LaAlO₃/SrTiO₃ sample and its microwave measurement setup. **a** LaAlO₃/SrTiO₃ sample inserted between the central strip and the ground of a CPW transmission line, in parallel with SMD inductors $L_1$ and resistors $R_1$. $C_p$ are protective capacitors that avoid dc current to flow through $L_1$ and $R_1$ without affecting $\omega_0$. **b** Sample circuit of impedance $Z_L$ in its microwave measurement set-up that includes an attenuated input line and an amplified readout line separated by a directional coupler. A bias-tee allows dc biasing of the sample. **c** Equivalent electrical circuit of the sample circuit including the SMDs and the LaAlO₃/SrTiO₃ heterostructure modeled by an impedance $Z_{2D}$ in parallel with a capacitor $C_{STO}$. The reflection coefficient $\Gamma(\omega)$, taken at the discontinuity between the CPW line and the sample circuit, is defined as the ratio of the complex amplitude of the reflected wave $A^{out}(\omega)$ to that of the incident wave $A^{in}(\omega)$

a metal whose Drude conductance is simply the inverse of the dc resistance (Fig. 2c). The circuit displays a resonance at frequency $\omega_0 = \frac{1}{\sqrt{L_1 C_{STO}}}$. When $\omega \approx \omega_0$, $Z_L$ becomes purely real and the microwave signal is dissipated in the sample circuit. As a result, an absorption dip is observed in $\Gamma(\omega)$ along with a $2\pi$ phase shift (Fig. 2b). $\omega_0$ varies upon gating because of the electric-field-dependent SrTiO₃ dielectric constant[27] (Fig. 2a). Thus, the deduced substrate capacitance, $C_{STO}$, decreases with the absolute value of the gate voltage (Fig. 2c). Note that $C_{STO}$ also includes a small contribution due to the circuit parasitic capacitance (≈3.5 pF) (see Methods section and Supplementary Fig. 2). According to the geometry of the sample, its value at $V_G = 0$ V corresponds to a dielectric dielectric constant $\epsilon_r = 23,700$ (Supplementary Fig. 3). In the superconducting state, the 2-DEG conductance acquires an imaginary part $G_2(\omega) = \frac{1}{L_k(T)\omega}$ that modifies $\omega_0$, since the total inductance of the circuit is now $\frac{L_1 L_k(T)}{L_1 + L_k(T)}$ ($L_1$ in parallel with $L_k(T)$). The superconducting transition observed in dc resistance for positive gate voltages, $V_G$, coincides with a shift of $\omega_0$ towards high frequency (Fig. 3b–d). We emphasize that this shift can already be detected in the uncalibrated $S_{21}(\omega)$ coefficient (Supplementary Fig. 4). In the absence of superconductivity (for $V_G < 0$ V), the resonance frequency remains unchanged as $C_{STO}$ has no temperature dependence in the range of interest (Fig. 3a). In this experiment, the typical microwave current flowing into the sample circuit is <5 nA, which is much lower than the critical current of the superconducting 2-DEG (≈5 μA).

**Superfluid stiffness and gap energy**. In the following, we determine the gate dependence of the important energy scales in

superconducting LaAlO₃/SrTiO₃ interfaces, and compare them with the BCS theory predictions. In Fig. 4a, we plot the gate dependence of the experimental superfluid stiffness $J_s^{exp} = \frac{\hbar^2}{4e^2 L_k}$ extracted from $L_k$ at the lowest temperature $T = 20$ mK (≈0 K in the following). On the same logarithmic scale, we also show the gate dependence of the superconducting $T_c$ defined as the temperature where $R_{dc} = 0$ Ω. The accuracy in the determination of the superfluid stiffness is limited by the uncertainty on the exact value of the circuit inductance $L_1$ and the contribution of the sample geometrical inductance. The total error, corresponding to the gray outline on Fig. 4a, is estimated to be lower than 15% for all gate voltages (Supplementary Note 2).

The superconducting 2-DEG is in the dirty limit in which the elastic scattering time $\tau$ is much shorter than the superconducting gap $\Delta(T = 0)$ ($\frac{\Delta(0)\tau}{\hbar} \simeq 5.5 \times 10^{-3}$). Within this limit and for $\omega \ll \Delta(0)/\hbar$, the zero-temperature superfluid stiffness of a single-band BCS superconductor can be expressed as a function of $\Delta(0)$[28]:

$$J_s(0) = \frac{\pi\hbar}{4e^2 R_n} \cdot \Delta(0), \qquad (2)$$

where $R_n = R_{dc}(450\ mK)$ is the normal state resistance (Fig. 2c) that accounts for the reduction of stiffness because of scattering.

$J_s^{exp}$ increases continuously with gate voltage in the entire phase diagram in agreement with a previous report[10]. Moreover, a remarkable agreement is obtained between experimental data ($J_s^{exp}$) and BCS prediction ($J_{BCS}$) in the overdoped (OD) regime defined by $V_G > V_G^{opt} \simeq 27$ V, assuming a gap energy $\Delta(0) = 1.76 k_B T_c$ in Eq. (2) (Fig. 4a). In this regime, the superfluid stiffness $J_s^{exp}$ takes a value much higher than $T_c$ in agreement with

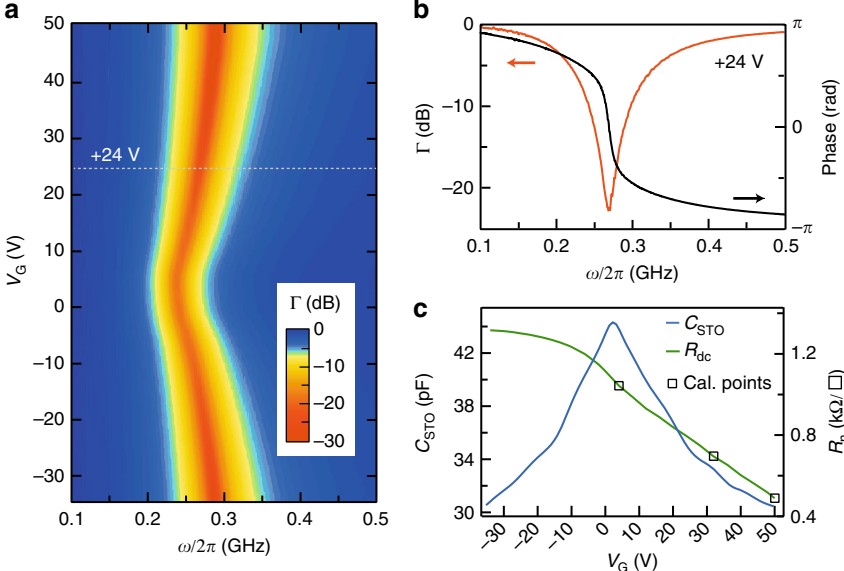

**Fig. 2** Resonance of the sample circuit in the normal state at $T = 450$ mK. **a** Magnitude of $\Gamma(\omega)$ in dB (color scale) as a function of $\omega$ and $V_G$. **b** Magnitude and phase of $\Gamma(\omega)$ at $V_G = +24$ V. **c** Capacitance $C_{STO}$ extracted from the resonance frequency $\omega_0 = 1/\sqrt{L_1 C_{STO}}$ (left axis) and normal dc resistance $R_n$ (right axis) as a function of $V_G$. Square symbols indicate the values of $V_G$ used for calibration

the BCS paradigm. However, in the underdoped (UD) regime, corresponding to $V_G < V_G^{opt}$, a discrepancy between the data and the BCS calculation is observed. The superfluid stiffness $J_s^{exp}$ drops significantly while $T_c$ and $J_{BCS}$ evolve smoothly before vanishing only when approaching closely the quantum critical point where $T_c \simeq 0$ K ($V_G = 4$ V). This indicates that the loss of phase coherence of the superconducting condensate is stronger than what expected taking into account conventional scattering by defects, as encoded in Eq. (2). Such a behavior can then be ascribed to strong phase fluctuations probably reinforced by the presence of spatial inhomogeneities which has been proposed as an explanation for the observed broadening of the super-conducting transitions[29,30]. In this context, it was shown that the 2-DEG in LaAlO₃/SrTiO₃ interfaces exhibits a behavior similar to the one of a Josephson junction array consisting of superconducting islands coupled through a metallic 2-DEG[11,31]. Whereas in the OD regime the islands are robust and tightly connected at $T \simeq 0$ K (homogeneous-like), in the UD regime, the charge carrier depletion makes the array more dilute. In this case, the system can maintain a rather high $T_c$ ($R_{dc} = 0$ Ω) as long as the dc current can follow a percolating path through islands. However, the macroscopic stiffness $J_s^{exp}$ is suppressed by phase fluctuations between islands and is therefore lower than that expected in a homogenous system of similar $T_c$.

We now convert $J_s^{exp}$ into a superconducting gap energy $\Delta_s^{exp}$ through Eq. (2) (Fig. 4b). Strikingly, these two characteristic energy scales of superconductivity evolve quite differently with doping. While $J_s^{exp}$ continuously increases with $V_G$ (Fig. 4a), $\Delta_s^{exp}$ has a dome-shaped dependence (Fig. 4b). More precisely, in the OD regime, $\Delta_s^{exp}$ coincides with $1.76 k_B T_c$, and decreases like $T_c$ while the superfluid stiffness increases: this is a clear indication that in this regime $T_c$ is controlled by the pairing energy as in the BCS scenario. The maximum energy gap at optimal doping ($V_G^{opt} \simeq 27$ V) is $\Delta_s^{exp} \simeq 23$ µeV. This is in agreement with the BCS gap identified recently by Stornaiuolo et al.[32] at optimal doping using spectroscopic Josephson junctions in LaAlO₃/SrTiO₃ interfaces of similar $T_c$. By using tunneling spectroscopy on planar Au/LaAlO₃/SrTiO₃ junctions, Richter et al.[9] have reported an energy gap in the density of states of $\simeq 40$ µeV for optimally doped LaAlO₃/SrTiO₃ interfaces of higher $T_c$. In spite of this

significantly higher gap energy, this corresponds to a $\frac{\Delta}{k_B T_c}$ ratio of 1.7, similar to our result. In the OD regime, we also checked that the gap value extracted from a BCS fit of the temperature dependence of $J_s^{exp}$ matches $\Delta_s^{exp}$ obtained by Eq. (2) (Supplementary Note 3 and Supplementary Fig. 6). In the UD regime of the phase diagram, $\Delta_s^{exp}$ departs from $1.76 k_B T_c$ which is in contradiction with BCS theory. This behavior is also different from that of the tunneling gap which was found to increase in the UD regime[9]. In addition, a pseudogap has been observed above $T_c$ in this regime, as also reported in high-$T_c$ superconducting cuprates[33,34] or in strongly disordered films of conventional superconductors[28,35,36]. The results obtained by the two experimental approaches can be reconciled by considering carefully the measured quantities. In our case, the superconducting gap $\Delta_s^{exp}$ probed by microwaves is directly converted from the stiffness of the superconducting condensate and is therefore only reflective of the presence of a true phase-coherent state. On the other hand, tunneling experiments probe the single particle density of states, and can evidence pairing even without phase coherence. The two experimental methods provide complementary informations which indicate that in the UD region of the phase diagram, the superconducting transition is dominated by phase coherence rather than by electron pairing. In this case, the energy gap cannot be extracted from Eq. (2), which is valid only for BCS superconductors. Notice that in the low carrier density region corresponding to $V_G < 0$, some non-connected superconducting islands could already form without contributing to the macroscopic stiffness of the 2-DEG. Recently, preformed electron pairs without phase coherence has also been evidenced in SrTiO₃-based nanostructures raising the question of a possible Bose–Einstein condensation mechanism where pairing precedes the formation of the superconducting state[37].

**Multiband transport.** A simplified scheme of the band structure in the interfacial quantum well is presented in Fig. 5a, b[38]. The degeneracy of the three $t_{2g}$ bands is lifted by the confinement in the $z$ direction, leading to a splitting that is inversely proportional to the effective masses $m_z$ along this direction. $d_{xy}$ subbands are isotropic in the interface plane with an effective mass $m_{xy} = 0.7 m_0$,

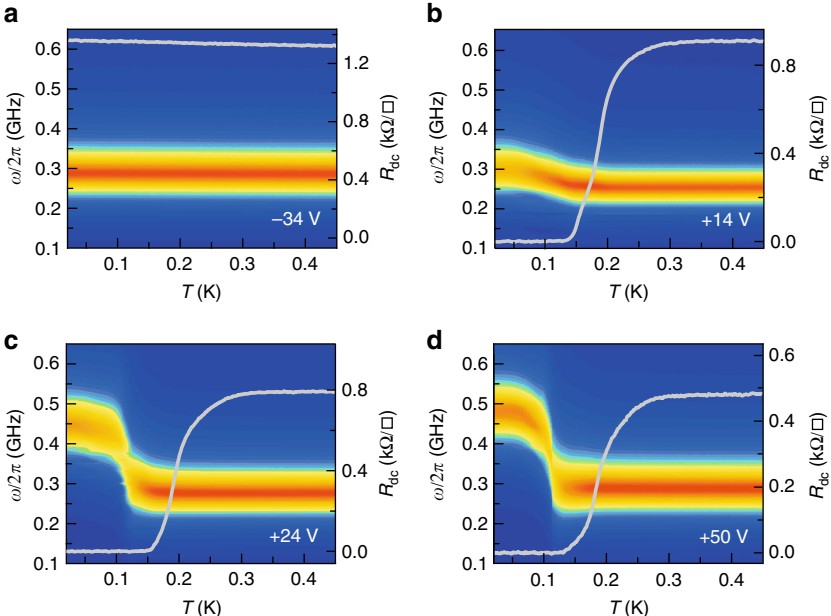

**Fig. 3** Resonance of the sample circuit in the superconducting state. Magnitude of $\Gamma(\omega)$ in dB (color scale) as a function of frequency and temperature for the selected gate values, $V_G = -34$ V (**a**), $V_G = +14$ V (**b**), $V_G = +24$ V (**c**), and $V_G = +50$ V (**d**). The corresponding dc resistance as a function of temperature is shown in gray solid lines (right axis)

whereas the $d_{xz}/d_{yz}$ bands are anisotropic with a corresponding average mass $m_{xz/yz} = \sqrt{m_x m_y} \simeq 3.13 m_0$. At low carrier densities, we expect several $d_{xy}$ subbands to be populated, whereas at higher density ($V_G > 0$ V), the Fermi energy should enter into the $d_{xz}/d_{yz}$ bands. Multiband transport in LaAlO$_3$/SrTiO$_3$ and LaTiO$_3$/SrTiO$_3$ interfaces has been observed experimentally in various magneto-transport experiments including quantum oscillations[39,40], magneto-conductance[15,41], and Hall effect[2,3,42–44]. Yang et al. [39] recently showed that, in addition to a majority of low-mobility carriers (LMCs), a small amount of high-mobility carriers (HMCs) is also present, with an effective mass close to the $m_{xz/yz}$ one. Despite a band mass substantially higher than the $m_{xy}$ one, these carriers acquire a high mobility since $d_{xz/yz}$ orbitals extend deeper in SrTiO$_3$ where they recover bulk-like properties, including reduced scattering, higher dielectric constant and better screening. In Hall effect measurements, the Hall voltage is linear in magnetic field $B$ in the low doping regime corresponding to one-band transport, but this is not the case at high doping because of the contribution of a new type of carriers (the HMC)[3]. We performed a two-band analysis of the Hall effect data combined with gate capacitance measurements to determine the contribution of the two populations of carriers to the total density $n_{tot}$ (Fig. 5c)[3]. The first clear signature of multiband transport is seen when the Hall carrier density $n_{Hall}$, measured in the limit $B \to 0$, drops with $V_G$ instead of following the charging curve of the gate capacitance ($n_{tot}$ in Fig. 5d). Figure 5d, e show that LMC of density $n_{LM}$ are always present, whereas a few HMC of density $n_{HM}$ are injected in the 2-DEG for positive $V_G$, which corresponds to the region of the phase diagram where superconductivity is observed. In consistency with quantum oscillations measurements, we identify the LMC and the HMC as coming from the $d_{xy}$ and $d_{xz}/d_{yz}$ bands, respectively, and we emphasize that the addition of HMC in the quantum well triggers superconductivity.

## Discussion

To further outline the relation between HMC and superconductivity, we extract the superfluid density $n_s^{2D}$ from $J_s^{exp}$

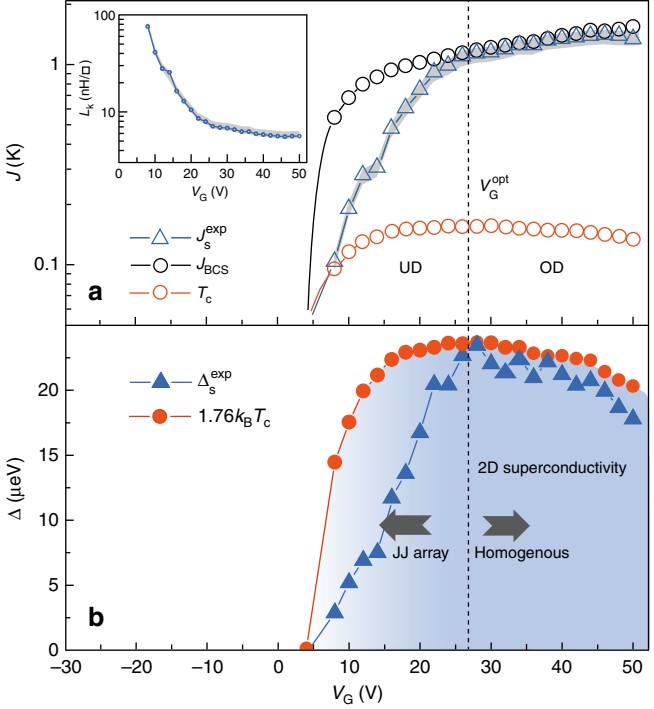

**Fig. 4** Superfluid stiffness and phase diagram. **a** Experimental superfluid stiffness $J_s^{exp}(T \simeq 0)$ (open triangles) as a function of $V_G$ compared with $T_c$ taken at $R_{dc} = 0$ $\Omega$ (red open circles), and with the BCS theoretical stiffness $J_{BCS}$ expected from Eq. (2) assuming $\Delta(0) = 1.76 k_B T_c$ (black open circles). The gray outline indicate the total error margin in the determination of $J_s^{exp}(T \simeq 0)$. Inset) $L_k(T \simeq 0)$ as a function of $V_G$ and error margin (gray outline). **b** Superfluid stiffness converted into a gap energy $\Delta_s^{exp}(T \simeq 0)$ as a function of $V_G$ (plain triangles) compared with the expected BCS gap energy $1.76 k_B T_c$ (plain circles)

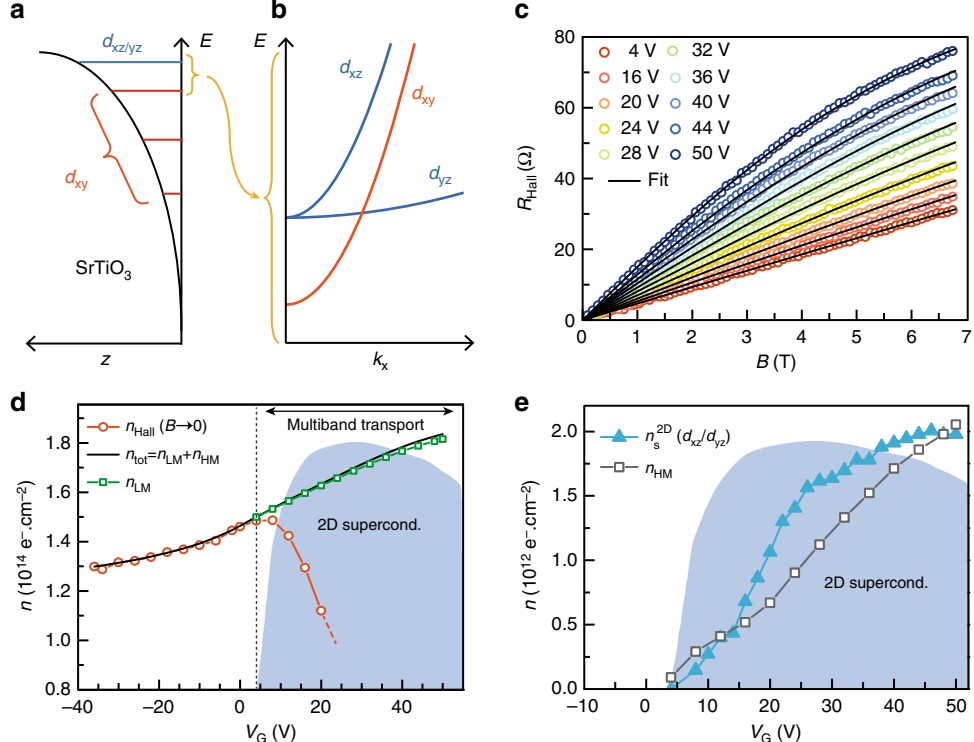

**Fig. 5** Superfluid density and Hall effect analysis. **a** Scheme of the interfacial quantum well showing the splitting of the $t_{2g}$ bands. **b** Simplified scheme of the band structure taking into account only the last filled $d_{xy}$ subband, the $d_{xz}$ band, and the $d_{yz}$ band. **c** Hall resistance as a function of magnetic field for different $V_G > 0$ (open circles), fitted by at two-band model (black solid lines) (see Methods). **d** Hall carrier density $n_{\text{Hall}} = \frac{B}{eR_{\text{Hall}}}$ extracted in the limit $B \to 0$ (red open circles) and LMC density $n_{\text{LM}}$ extracted from the two-band analysis (green open squares). The total carrier density $n_{\text{tot}}$ is obtained by matching the charging curves of the gate capacitance with $n_{\text{Hall}}$ at negative $V_G$ (black solid line). The unscaled $T_c$ dome in the background indicates the region where superconductivity is observed. **e** Superfluid density $n_s^{\text{2D}}$ calculated from $J_s^{\text{exp}}$ using a mass $m_{xz/yz}$ (plain triangles), compared with the HMC density $n_{\text{HM}}$ (open squares)

assuming a mass $m_{xz/yz}$ for the electrons, and plot it as a function of the gate voltage (Fig. 5e). It increases continuously to reach $n_s^{\text{2D}} \simeq 2 \times 10^{12} \ e^- \text{cm}^{-2}$ at maximum doping, which is approximately 1% of the total carrier density. The comparison of $n_s^{\text{2D}}$ with $n_{\text{HM}}$ shows that, unexpectedly, both quantities have a very similar dependence with the gate voltage and almost coincide numerically (Fig. 5e). This suggests that the emergence of the superconducting phase is mainly related to the filling of $d_{xz}/d_{yz}$ bands, whose high density of states is favorable to superconductivity. Nevertheless, in the presence of interband coupling, superconductivity may also be induced in some $d_{xy}$ subbands which would then slightly contribute to the total superfluid density.

Bert et al.[10] measured the superfluid density in LaAlO₃/SrTiO₃ interfaces using a scanning SQUID technique. The overall gate dependence is similar in both experiments, including in the OD regime where the superfluid density keeps increasing while $T_c$ is reduced. However, in our case $n_s^{\text{2D}}$ is lower despite a much higher carrier density ($n \simeq 1.8 \times 10^{14} \ e^- \text{cm}^{-2}$ at maximum doping) which corresponds to the upper limit of the doping range commonly observed in LaAlO₃/SrTiO₃ interfaces. The fact that $n_s^{\text{2D}} \simeq n_{\text{HM}}$ may be somewhat intriguing as the dirty limit that we used in Eq. (2) implies that $n_s^{\text{2D}}$ should correspond to a fraction of the total normal carrier density and not to $n_{\text{HM}}$. To clarify this situation, it is needed to go beyond single-band superconductor models that cannot account correctly for the unusual $t_{2g}$-based interfacial band structure of LaAlO₃/SrTiO₃ interfaces. Further investigations of recent experimental[45] and theoretical[46] developments on superconductors having two dissimilar bands (e.g., clean and

dirty, weak, and strong coupling) should provide the starting framework to address this question.

In summary, we have measured the superfluid stiffness $J_s$ of LaAlO₃/SrTiO₃ interfaces by implementing a resonant microwave transport experiment. Whereas a good agreement with the BCS theory is observed at high carrier doping, we find that the suppression of $T_c$ at low doping is controlled by the loss of macroscopic phase coherence instead of electron pairing strength as in standard BCS theory. The corresponding superfluid density represents only a small fraction of the total electrons density. We emphasize here that the monotonic raise of $n_s^{\text{2D}}$ with gate voltage indicates that the decrease of $T_c$ in the OD region of the phase diagram cannot be attributed to a loss of superfluid density. The gate dependence of $n_s^{\text{2D}}$ agrees qualitatively with the density of HMCs extracted from multiband Hall effect. We therefore propose that the emergence of superconductivity upon gating is related to the weak filling of the $d_{xz}/d_{yz}$ bands taking place at higher energy in the quantum well. In addition to having a larger density of states, these $d_{xz}/d_{yz}$ bands also extends much deeper in the substrate due to their out-of-plane mass. Away from the interface, the dielectric constant, which is most probably a fundamental ingredient for electron pairing[47], is less affected by the interfacial electric field and therefore closer to its nominal value. These delocalized electrons therefore recover properties similar to the ones found in bulk SrTiO₃, including BCS superconductivity[48]. Our finding is consistent with the observation of a gate-independent superconductivity in (110)-oriented LaAlO₃/SrTiO₃ interfaces for which the $d_{xz}/d_{yz}$ bands have a lower energy than the $d_{xy}$ subbands and are therefore always filled[14].

## Methods

**Sample growth**. In this study, we used 8-uc-thick LaAlO$_3$ epitaxial layers grown on $3 \times 3$ mm$^2$ TiO$_2$-terminated (001) SrTiO$_3$ single crystals by pulsed laser deposition. The substrates were treated with buffered hydrofluoric acid to expose TiO$_2$-terminated surface. Before deposition, the substrate was heated to 830 °C for 1 h in an oxygen pressure of $7.4 \times 10^{-2}$ mbar. The thin film was deposited at 800 °C in an oxygen partial pressure of $1 \times 10^{-4}$ mbar. The LaAlO$_3$ target was ablated with a KrF excimer laser at a rate of 1 Hz with an energy density of 0.56–0.65 J cm$^{-2}$. The film growth mode and thickness were monitored using reflection high-energy electron diffraction (STAIB, 35 keV) during deposition. After the growth, a weakly conducting metallic back-gate of resistance ~100 kΩ (to avoid microwave shortcut of the 2-DEG) is deposited on the backside of the 200-μm-thick SrTiO$_3$ substrate.

**Calibration procedure**. In this experiment, the resonance frequency shift and correspondingly $J_s$ can be extracted directly from the raw measurements of $S_{21}$ in most of the regions of the phase diagram (Supplementary Fig. 4). Nevertheless, a calibration procedure can be applied to relate $S_{21}$ measured with the Vector Network Analyzer to the reflection coefficient $\Gamma = (Z_L - Z_0)/(Z_L + Z_0)$ of the sample circuit. This procedure also suppresses parasitic signals mainly due to wave interferences in the microwave setup, and improves the precision on the measurement as illustrated in Supplementary Fig. 4b. The microwave setup can be modeled using the scattering matrix formalism as shown in Supplementary Fig. 5. The relation between the transmission coefficient between port 1 and port 2 $S_{21}(\omega)$ and the reflection coefficient of the sample circuit $\Gamma(\omega)$ is given by

$$S_{21} = \gamma + \frac{\alpha' \Gamma}{1 - \delta \Gamma}, \tag{3}$$

where $\alpha' = \alpha\beta$, $\gamma$, and $\delta$ are three error complex coefficients. They can be determined using three known values of $\Gamma = (Z_L - Z_0)/(Z_L + Z_0)$ which are obtained by imposing three different impedances $Z_L$. It is customary to use an open, a short, and a matched load as standard impedances to calibrate microwave setup. However, such method is neither adapted to our very low temperature experiment nor to our sample circuit geometry. Instead, our setup was calibrated by directly varying the impedance $Z_L$ of the sample circuit with gate value. The main advantage of this method is that it fully takes into account the local microwave environment of the sample. The gate controls both the normal resistance of the 2-DEG whose value can be measured in dc and $C_{STO}$ which can be extracted from $\omega_0$. In practice, we choose a set of three gate values which correspond to well-separated resonance frequencies. Other sets allow the accuracy of the calibration to be checked. Supplementary Fig. 4 shows a comparison between the raw measurement of $S_{21}$ and the corresponding calibrated $\Gamma$ coefficient both in the normal state and the superconducting state for $V_G = 50$ V.

**SrTiO$_3$ dielectric constant**. The dielectric constant of the SrTiO$_3$ substrate can be retrieved from the value of $C_{STO}$ plotted in Fig. 2c of the main text. For that, we performed numerical simulation using finite element method. We consider a 200-μm-thick $3 \times 3$ SrTiO$_3$ substrate covered by two 100-μm-wide Au/Ti strips as represented in Supplementary Fig. 1. Supplementary Fig. 4a shows the distribution of the electrostatic potential when one volt is applied on one Au/Ti strip while the other one is at the ground. Arrows indicate the direction of the electric field. The numerical simulation provides the corresponding capacitance between the two Au/Ti strips for a given dielectric constant $\epsilon_r$. The gate dependence of $\epsilon_r$ that corresponds to the value of $(C_{STO} - C_{para})$ measured experimentally is shown in Supplementary Fig. 4b. At $V_G = 0$ V, $\epsilon_r \simeq 23,700$, which is consistent with the value found in the literature[27].

**Multiband Hall effect and gate capacitance**. The dependence of the total carrier density $n_{tot}$ with $V_G$ is obtained by integrating the gate capacitance $C(V_G)$, measured by standard lock-in technique, over the gate voltage range

$$n_{tot}(V_G) = n_{tot}(V_G = -36 \text{ V}) + \frac{1}{eA} \int_{-36}^{V_G} C(V) \mathrm{d}V, \tag{4}$$

where $A$ is the area of the sample and $n_{tot}(V_G = -36$ V$)$ is matched to $n_{Hall}$ since in this low doping regime the Hall effect is linear in magnetic field (single-band transport). In the multiband transport regime corresponding to $V_G > 0$, the Hall resistance has been fitted with a two-band model

$$R_{Hall} = \frac{B}{e} \frac{\frac{n_1 \mu_1^2}{1 + \mu_1^2 B^2} + \frac{n_2 \mu_2^2}{1 + \mu_2^2 B^2}}{\left[\frac{n_1 \mu_1}{1 + \mu_1^2 B^2} + \frac{n_2 \mu_2}{1 + \mu_2^2 B^2}\right]^2 + \left[\frac{n_1 \mu_1^2 B}{1 + \mu_1^2 B^2} + \frac{n_2 \mu_2^2 B}{1 + \mu_2^2 B^2}\right]^2}, \tag{5}$$

where $n_1$ and $n_2$ are the 2D electron densities and, $\mu_1$ and $\mu_2$ the corresponding mobilities. Fits are performed with the two constraints: $n_{tot} = n_1 + n_2$ and $1/R_{dc} = e(n_1 \mu_1 + n_2 \mu_2)$. The two populations of electrons are then identified as the LMC and the HMC.

**Data availability**. All data that support the findings of this study are available from the corresponding authors upon request.

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

## Acknowledgements

We acknowledge R. Lobo, C. Castellani, and J. Lorenzana for useful discussions. This work has been supported by the Région Ile-de-France in the framework of CNano IdF, OXYMORE, and Sesame programs, by CNRS through a PICS program (S2S) and ANR JCJC (Nano-SO2DEG). L.B. acknowledges financial support by the Italian MAECI under the Italian-India collaborative project SUPERTOP-PGR04879. Part of this work has been supported by the IFCPAR French-Indian program (contract 4704-A). Research in India was funded by the CSIR and DST, Government of India.

## Author contributions

G. S. and A. J. performed the measurements assisted by N.B. Samples were fabricated by P.K. and E.L. under the supervision of A.D., R.C.B., A.B., and M.B. G.S., A.J., and N.B. carried out the analysis of the results and wrote the article with the help of L.B., M.B., and J.L. F.C., C.F.-P., A.B., M.G., and S.C. contributed to discussions of the results.

## Additional information

**Competing interests:** The authors declare no competing financial interests.

