## [Peer Review File · Nature Communications]

Reviewers' comments:

Reviewer #1 (Remarks to the Author):

The manuscript "Competition between electron pairing and phase coherence in superconducting interfaces" by G. Singh et al. reports an experimental study of the superconducting stiffness J_s and of the superfluid density as function of the carrier density of the 2DEG formed at the LAO/STO interface.

The data are measured at very low temperatures using a microwave coplanar waveguide in which the LAO/STO sample is embedded.

Using some assumptions, the Authors determine the carrier density dependence of the gap from the kinetic inductance of the 2DEG and compare it to the BCS gap determined from the superconducting T_c . The total carrier density is determined from the CSTO capacitance and total area of sample, taking into account the non-linear dielectric constant of STO. Moreover, the Authors compare the total carrier density obtained by this method and the Hall effect conductivity, which is fitted using a two band model in the normal state, from which it is possible to determine the fraction of $3d_{xy}$ (low mobility) and $3d_{xz,yz}$ (high mobility) carriers in the system. The superfluid stiffness is then converted in a superfluid density n_s assuming an effective mass equal to the average effective mass of $3d_{xz,yz}$ carriers, and is then compared to the carrier density of the two bands as function of the gate voltage. The Authors find a coincidence between the filling of the $3d_{xz,yz}$ band, the superfluid density and the emergence of the superconductivity in the system.

There are two main messages in the paper:

The Authors find that, while in the overdoped region of the phase diagram the gap (determined from J_s) follows the BCS expectations, in the underdoped region J_s decreases much faster than the BCS value with T_c , Δ and with the carrier density. This result is interpreted as a proof of a loss of phase coherence, possibly related to the formation of SC islands Josephson-coupled in the underdoped region of the phase diagram, instead of the usual BCS scenario where superconductivity appears as soon as Cooper pairs are formed. This interpretation has been already anticipated by the Authors in some works from the analysis of the dc R vs T transition as function of the magnetic field and of the gate voltage, thus it is not new.

Superconductivity in the LAO/STO system is exclusively related to $3d_{xz,yz}$ carriers. This result has been anticipated in different papers from the Geneva group [Phys. Rev. Lett. 104, 126803 (2010)], however the possible coincidence between the occurrence of SC and the filling of $3d_{xz,yz}$ carriers has not been firmly established yet. This paper could give more compelling proofs of a SC state emerging from only $3d_{xz,yz}$ carriers in the LAO/STO system.

Overall the paper is well written and the results could effectively clarify some of the open issues in the field of the LAO/STO superconductivity. What is relatively new is the systematic measurement of the superfluid stiffness, which is relatively complex.

However, in my view, the paper should be revised before I can recommend publications for several reasons I list below.

-Remarks related to the method of measurement and its description:

a) The measurement of the microwave conductivity in LAO/STO is not straightforward at all. In particular, while few details are given for what concern the experimental method, all the possible source of errors in the estimation of the absolute value of L_k are not considered/explained. For example, the Authors use a CPW SMD circuit, and they replace one part of the transmission line with the sample. There are no details about the characteristics of the CPW device used, and about the effective coupling (how the CPW is connected to the LAO/STO?) between the CPW and the $3 \times 3 \times 0.1$ mm² square LAO/STO sample. These details are important. For example, the Authors assume that the main contribution to the resonant frequency is associated to the STO capacitance.

Is this reasonable? are they sure that the coupling capacitance does not contribute? Have they made any kind of test, replacing the non standard LAO/STO superconductor with some well-known BCS superconductor deposited on the same kind of STO substrate, like Nb or Al? The reasons why these issues are relevant is that a quantitative determination of the kinetic inductance and of the superfluid stiffness relies on the possibility to directly correlate the absolute value of the resonant frequency to the kinetic inductance in the superconducting state. This requires a calibration, and the precise knowledge of the total capacitance at each temperature and each gate voltage. About the calibration, a standard calibration of a spectrum analyzer is not sufficient. The Authors used the Z impedance of the system including the LAO/STO in its normal state at each gate voltage. It is unclear to me how the Authors performed this calibration and the details of it. For sure, a test using a standard SC is a much more reliable way to verify that the measurement system is able to determine the absolute value of the kinetic inductance of a BCS superconductor. If the error calibration is not done properly, then what can be obtained is just the the frequency shift respect the normal state value. Then this does not give an absolute value of the superfluid stiffness at zero temperature. However, it is possible to get in any case a good estimate of the superfluid stiffness from a fitting of the $\Delta f(T)$ data in the SC state, which is customary in microwave resonating methods to measure the absolute value of the penetration depth of a superconductor. This point, which is related to the above two issues, needs to be clarified.

b) Since the Authors uses the Sheet resistance in the normal state measured in dc, it would be important to know what is the reason why they expect that this quantity is not frequency dependent. At the microwaves, the losses are determined by the microwave surface resistance, which could coincides with the sheet resistance only if the are not additional scattering terms (due to disorder for example) which contribute.

c) Is the T_c used in the paper to calculate the BCS gap determined from the Resistive dc measurement? Is it the zero resistance T_c ? Are these measurements performed on the same device with the microwaves applied to the sample or not?

Thus in conclusion about the experimental method, my suggestion is to make all the possible experimental and additional checks (for example fitting of the $\Delta f(T)$ curve using a phenomenological two fluid model) to validate the absolute values of the superfluid stiffness. These additional results and details on the experimental technique should be reported in a supplementary material file.

-Remarks about the main messages given:

d) A loss of phase coherence is only one of the possible explanation for a reduction of the superfluid stiffness respect the BCS value in a superconductor. It is well known that the penetration depth measured at the microwaves can vary a lot even in standard superconductors due to extrinsic contribution, related to defects, like point, linear or other kind of defects. Additionally the non-linear dielectric constant of STO, usually by itself give rise to a shift of the resonant frequency (also measured by the Authors) which can be sample dependent. Thus the Author should explain why other scenarios are not plausible to explain the data, or smooth out their conclusion about this part of the paper, which is not, in my opinion, the most original and important one. The idea that in the underdoped phase LAO/STO is essentially composed by a network of 2D-SC islands Josephson coupled has been already anticipated by the Authors in previous publications. In the LAO/STO case, however, there are several studies on nano bridges [Appl. Phys. Lett. 101, 222601 (2012), Nano Lett. 15, 2627 (2015).], suggesting that the LAO/STO 2DEG is quite homogeneous at the nanoscales. How the Authors explain this contradiction with other data present in literature?

e) The gap and the superfluid density are estimated by making the assumption that LAO/STO is dirty single-gap BCS superconductor. Some of these assumptions are not fully justified. Recent works show that SC in LAO/STO could be rather unconventional [Phys. Rev. B 95, 140502 (2017), Phys. Rev. B 96, 014513 (2017)]. In my opinion, the determination of the gap from the superfluid stiffness is only an indirect indication of a single, BCS, gap. In general, the present study is sensitive to the main pairing channel, i.e. the pairing channel which carries most of the

superfluid density contribution. If another channel is present, maybe just in some fraction, and it is unconventional, it is clear that it could be much more sensitive to the experimental conditions/methods. For example microwave radiation could destroy any contribution to σ_2 of an unconventional order parameter, while being able to measure the major contribution to the superfluid stiffness due to the conventional channel.

f) The gap obtained at optimal doping in the present work is in better agreement with spectroscopic data in ref. [Phys. Rev. B 95, 140502 (2017)] (for one of the two gap found, the BCS one), but is half of the valued reported by Mannhart et al. This discrepancy should be discussed.

e) Fig. 5d and Fig. 5e reports the density of $3d_{xy}$ and $3d_{xz,yz}$ carriers and the superfluid density estimated from the microwave measurements. There is very likely an error in the vertical scale (a factor 10?) of Fig. 5d.

g) Still in Fig. 5e the superfluid density and the density of high mobility carriers are of the same order of magnitude but are different. In particular, it seems that there are other carriers contributing to superfluid density. Which carriers? Is moreover correct to assume a constant effective mass in the calculation of the superfluid density? Some care is needed about that.

h) Why the T_c seems different in the dc-resistance measurement and in the microwave measurement?

i) The temperature dependence of the superfluid stiffness is not discussed and not fitted by any model. Why? Additionally I notice some non monotonous behavior close to T_c in Fig. 3.

Reviewer #2 (Remarks to the Author):

Referee report on manuscript Nature Communications manuscript NCOMMS-17-16538A-Z by G. Singh et al.

In this paper, the authors study superconductivity in the now famous $\text{LaAlO}_3/\text{SrTiO}_3$ system – the interface between two band insulators that was found to be conducting and superconducting. The study focuses on the superconducting state and on the phase diagram of the system, T_c versus doping, the latter being controlled by an electrostatic gate. The authors use an original approach – a resonant microwave technique – that allows the kinetic inductance of the superconductor to be measured as a function of the gate voltage. From the kinetic inductance, the authors calculate the superfluid stiffness related to the superfluid density n_s and the superconducting gap. The authors find that the superfluid stiffness corresponds to the value expected in a BCS model (with $1.76k_B T_c$ equal to the gap) in the overdoped part of the phase diagram – from optimally doped samples to the most overdoped ones. In the underdoped part of the phase diagram, the superfluid stiffness is markedly reduced suggesting a T_c controlled by phase fluctuations and a behavior similar to the one of Josephson junction arrays. The authors further connect the very low superfluid density observed to superconducting carriers originating from the heavier – high mobility - d_{xz} , d_{yz} bands that are progressively filled at relatively large doping – the lowest (sub)bands being light d_{xy} ones.

This is a nice and interesting paper on a hot topic. The authors use an interesting technique to determine the superfluid stiffness and address the physics of the underdoped regime of the $\text{LaAlO}_3/\text{SrTiO}_3$ system. I think that the paper can be considered for Nature communications if the authors take into account my comments and address some important issues discussed below.

Comments:

-There are important references that are either missing or that are not discussed in sufficient details.

On the measurements of the superfluid density, the work of K. Moler is cited as reference 7 but

does not appear anywhere in the text.

Reference 10 - Bert et al. PRB 86 060503(R) (2012) must be discussed in some details since these authors measure precisely the superfluid density versus gate as in this paper. The authors should then stress what is different and what is new in their contribution?

On the question of the orbital ordering, reference 12 should be the paper of Saluzzo and co-workers (PRL 102, 166804 (2009)) since it is the first experimental report on the orbital reconstruction. The authors may want to cite Berner et al. too.

On the high mobility / low mobility carriers, several papers should be cited:

Bell et al. PRL 103, 226802 (2009)

Ben Shalom et al. PRL 105, 206401 (2010)

Fete et al. PRB 86, 201105(R) (2012)

Joshua et al. (ref. 14) Nat. Comm. (2012)

References 36 and 37 are not on the LaAlO₃/SrTiO₃ system.

On the role of the dx_z, dy_z bands on superconductivity, Gariglio et al. (APL Mater. 4, 060701 (2016)) discuss in detail the superconducting phase diagram of the system, the key role of the heavy high-density of states dx_z, dy_z bands for superconductivity and the fact that this could explain the low superfluid density observed by Bert et al.

On the theory side, Valentini et al. (ArXiv 1611.07763) propose a scenario that describes the phase diagram of the LaAlO₃/SrTiO₃ system and explains the role of the heavy subbands.

-p.2 - The authors say that they propose that the filling of the dx_z, dy_z bands controls the emergence of superconductivity. As mentioned just above, this scenario has been proposed before by several authors. The text should thus be rephrased highlighting the fact that, in this paper, the carrier density in the dx_z, dy_z bands is compared to the superfluid density making this statement much stronger.

-p.2 - The authors say that they extract the superfluid stiffness and the superconducting gap from their measurements. As I understand, they determine the kinetic inductance of the superconductor as a function of the doping and from this calculate (in a BCS model) the gap – they do not measure the gap independently as one would do in a tunneling experiment. This point should be clarified.

-The technique used to determine L_k is very interesting. Figure 1 however does not allow the reader to understand the experimental set-up and the measurements. Improving the sketch of the device would be useful as well as a better description of the measurements themselves.

-From Figure 2.c, one can probably extract the dielectric constant of SrTiO₃ – it would be interesting to plot the latter as a function of the gate.

-An important comment concerns Fig. 4 and the discussion of the data that I find confusing. The equations that are used: the BCS gap calculated from T_c and the relation between the superfluid stiffness and the gap are valid for a BCS superconductor (for a mean field transition). If one is having a Kosterlitz-Thouless like transition – a transition controlled by phase fluctuations, the establishment of phase coherence is not linked to the gap and one thus does not get the gap from the phase stiffness – the two equations are thus valid in the overdoped regime but not in the underdoped one.

Now, I think I understand what the authors want to do; they want to demonstrate that the underdoped part of the phase diagram is not BCS-like. Fine, but the way the data are presented is highly confusing – Fig. 4 shows Δ_{BCS} versus gate – at the QCP, Δ_{BCS} obviously goes to zero since (the establishment of phase coherence) T_c goes to zero (as well as Δ_{exp} – making a comparison with the tunneling data (p.6) irrelevant). At the QCP, however, the mean field T_c

and the pairing gap may well be finite – very large if one is believing the Stuttgart tunneling data – whatever the behavior of the gap is, the Figure is very misleading for the reader since one is seeing Δ_{BCS} versus gate as the equation used is not valid in the underdoped part of the phase diagram.

-What about a Kosterlitz-Thouless transition? This point is essentially not discussed as the authors mention phase fluctuations and the fact that the system behaves as a Josephson junction array whose physics is the one of a 2D xy model. This issue should be discussed in detail keeping in mind that, although disorder and/or inhomogeneities in the superfluid density may wash out the signatures of a BKT transition, such a transition may take place.

-Some less important points:

p.1 the authors mention the superconducting dome ending into a QCP – there are probably two QCP's (at both endpoints).

p.5 ... with a 2π shift (Fig.2b) – not 3π .

Response to referee A

We thank the referee for his/her critical reading of our manuscript and suggestions of improvement. We detail below our responses to his/her comments, together with the changes we made on the revised version of the article. We hope that this new version will be suitable for publication.

I. Remarks related to the method of measurement and its description

The referee raised several issues regarding the experimental method used to measure the superfluid density. We agree that technical details were missing in the first version of the manuscript. As the measurement method is new, we understand that the reader may be interested in having more practical information. For this reason, we have included a supplementary material devoted to this aspect in the revised submission.

Before addressing point by point the specific questions of the referee, we would like to summarize the main advantages of our method, which is the result of a long experimental development.

1) The principle of the measurement is conceptually very simple. The extraction of $J_s(T)$ relies on the measurement of the kinetic inductance L_k which is determined through the resonance frequency $f_0 = \frac{1}{2\pi\sqrt{L_{tot}C_{tot}}}$.

The resonant sample circuit **has been designed such that L_1 and L_K dominate the total inductance**. L_1 is an SMD microwave inductor whose value has a negligible temperature dependence. In the normal state $L_{tot} = L_1$ and c_{tot} is determined with a very good accuracy from f_0 . In the superconducting state, $L_{tot} = \frac{L_1 L_k}{L_1 + L_k}$ and f_0 gives a direct access to L_K .

As we will see, c_{tot} is dominated by the SrTiO₃ capacitance (C_{sto}) but this is not particularly relevant to extract L_K .

2) The **resonance frequency doesn't depend on the resistive part** of the circuit. This latter is a combination of the SMD resistor $R_1=70 \Omega$ in parallel with the sample resistance R and a weak contribution due to the losses of the STO substrate. An error or an uncertainty on the determination of any of these resistive contributions, including the characteristic impedance of the CPW line Z_0 , will not affect the determination of $J_s(T)$. Nevertheless, the total resistance of the circuit needs to be close to $Z_0=50 \Omega$ to make the resonance visible. If needed, the value of the total resistance can be extracted from the absorption dip at the resonance (amplitude and width).

3) The resonance frequency **can be measured directly from raw data without calibration**.

Thanks to this resonant microwave transport method we have been able to extract reliably $J_s(T)$ over the entire phase diagram and to compare it quantitatively with BCS predictions.

We now address the specific questions of the referee.

Referee's comment a): *The measurement of the microwave conductivity in LAO/STO is not straightforward at all. In particular, while few details are given for what concern the experimental method, all the possible source of errors in the estimation of the absolute value of L_k are not considered/explained. For example, the Authors use a CPW SMD circuit, and they replace one part of the transmission line with the sample. There are no details about the characteristics of the CPW device used, and about the effective coupling (how the CPW is connected to the LAO/STO?) between the CPW and the $3 \times 3 \times 0.1 \text{ mm}^2$ square LAO/STO sample. These details are important. For example, the Authors assume that the main contribution to the resonant frequency is associated to the STO capacitance. Is this reasonable? are they sure that the coupling capacitance does not contribute? Have they made any kind of test, replacing the non standard LAO/STO superconductor with some well-known BCS superconductor deposited on the same kind of STO substrate, like Nb or Al? The reasons why these issues are relevant is that a quantitative determination of the kinetic inductance and of the superfluid stiffness relies on the possibility to directly correlate the absolute value of the resonant frequency to the kinetic inductance in the superconducting state. This requires a calibration, and the precise knowledge of the total capacitance at each temperature and each gate voltage. About the calibration, a standard calibration of a spectrum analyzer is not sufficient. The Authors used the Z impedance of the system including the LAO/STO in its normal state at each gate voltage. It is unclear to me how the Authors performed this calibration and the details of it. For sure, a test using a standard SC is a much more reliable way to verify that the measurement system is able to determine the absolute value of the kinetic inductance of a BCS superconductor. If the error calibration is not done properly, then what can be obtained is just the the frequency shift respect the normal state value. Then this does not give an absolute value of the superfluid stiffness at zero temperature. However, it is possible to get in any case a good estimate of the superfluid stiffness from a fitting of the $\Delta f(T)$ data in the SC state, which is customary in microwave resonating methods to measure the absolute value of the penetration depth of a superconductor. This point, which is related to the above two issues, needs to be clarified.*

CPW line

Our answer : the sample circuit is designed in a $t=50$ micrometers thick copper layer covering a $h=1.3$ mm thick dielectric material (ceramic-filled PTFE composites) having a dielectric constant of 10.5 at low temperature (temperature independent). The CPW transmission line consists in a $s=3$ mm wide central line separated from the ground by a gap of $W=1.1$ mm. Its characteristic impedance can be derived using conformal mapping techniques in the limit where $t \rightarrow 0$ ¹.

$$Z_0 = \frac{30\pi}{\sqrt{\epsilon_{re}}} \frac{K(k_1')}{K(k_1)}$$

where K and K' represent the complete elliptic integral of the first kind, $k_1 = \frac{s}{s+2W}$ and

$$k_1' = \sqrt{1-k_1^2}$$

ϵ_{re} is the effective dielectric constant given by

$$\epsilon_{re} = 1 + \frac{\epsilon_r - 1}{2} \frac{K(k_2)}{K'(k_2)} \frac{K'(k_1)}{K(k_1)}$$

where $\epsilon_r=10.5$ is the dielectric constant of the dielectric material, $k_2 = \frac{\sinh(\pi s / 4h)}{\sinh(\pi(W + s / 2) / 2h)}$

$$\text{and } k_2' = \sqrt{1-k_2^2}$$

¹ Microstrip Lines and Slotlines, second Edition K.C Gupta ARTECH HOUSE, BOSTON 1996

Corrections due to the finite thickness of the metallic layer which tend to lower Z_0 have been introduced in the book of Gupta et al¹. This results in a $Z_0=50 \pm 1 \Omega$ due to a weak uncertainty on the exact geometry of the CPW line introduced during the fabrication. Alternatively, some calculators are available on the web (see for instance <http://janielelectronics.com/szamitasok/TransmissionLine/CoplanarWaveguideCalculator/janilab.php>)

The value of Z_0 was checked experimentally by measuring the reflection and transmission coefficient of a CPW line of the same geometry.

In our experiment, the CPW line is soldered to a SMA microwave connector, which is in turn connected to a semi-rigid coaxial cable. We emphasize here that an uncertainty on the value of Z_0 **will not affect the resonance frequency and the determination of J_s** . Indeed, as Z_0 is real, an error will only affect the amplitude of the resonance dip.

Changes : Details on the CPW line geometry were added in the Supplementary Material part I.

Sample

Our answer : A close-up view of the sample environment is shown in figure R1. After the growth, a weakly conducting metallic back-gate is deposited on the backside of the substrate. Its very high resistance ($\sim 100 \text{ k}\Omega$) avoids any short-cut of the microwave signal by the gate. Two $100 \mu\text{m}$ wide and 3 mm long Ti/Au strips are deposited at the edges of the sample to facilitate electrical contacts and increase the coupling. The sample is glued with GE Varnish on a 1 mm thick MgO substrate to isolate the gate from the electric ground of the cryostat. A section of the CPW is removed and the LAO/STO sample is inserted between the central line and the ground of the CPW. Multiple wire-bonds to the Ti/Au stripes covered by silver epoxy ensure a negligible impedance contacts between the 2-DEG and the CPW line both at DC and RF frequency ($<0.1\Omega$). A wire-bond allows connecting the gate with a tiny Ag contact on the MgO substrate.

Figure R1: close-up view of the sample inserted into the CPW line

Changes : *Figure R1 presenting a close-up view on the sample as well a paragraph on the sample environment were added in the Supplementary Material.*

SMD devices

Our answer : Our SMD microwave inductor L_1 and resistor R_1 have been extensively tested at low temperature to check their reliability. In particular, we made RLC circuits by replacing the LAO/STO sample with a SMD capacitor. Figure R2 shows the resonance frequency of such RLC circuit as a function of frequency. f_0 measured experimentally corresponds to the calculated value within 5% which corresponds to the tolerance factor on the nominal value of capacitance and inductance. It varies by less than 3% over the entire temperature range 20mK-300K and by less than 0.1% below 1K. Moreover this weak variation of the f_0 is probably entirely due to the SMD capacitor, which is not used in the true experiment (L_1 is controlled by the geometrical inductance of the coil which does not change with temperature). Four points DC measurements also confirm that the resistance R_1 doesn't change with temperature (<0.5%). Note that as already mentioned, an uncertainty on the value of R_1 will not affect the resonance frequency and the determination of J_s .

Figure R2: resonance frequency of a RLC circuit made with SMD devices ($L=14.5\text{nH}$ and $C=24\text{pF}$) as a function of temperature. The nominal resonance frequency is $f_0=269.8$ MHz and the measured one at $T=0$ is $f_0=274.7$ MHz.

Changes : A paragraph on the SMD devices was added in the Supplementary Material part I. Figure R2 has also been added.

Sources of errors.

Our answer : The sample circuit has been designed to minimize the contribution of the parasitic capacitance and parasitic inductance to the circuit. The dominant source of error on the determination of L_k is coming from the uncertainty on the value of the total inductance of the circuit, which we assume is L_1 in the normal state. Parasitic capacitance is not a source of error because we extract the total capacitance of the circuit (including the parasitic contribution) for each gate voltage in the normal state and because it is temperature independent in the range of interest.

Inductance of the circuit. The main contributions to the total inductance of the circuit are L_1 (SMD) and L_K (kinetic inductance of the superconducting 2-DEG). $L_1 = 10\text{nH}$ is a

known with a tolerance of 5% (according to manufacturer). In addition a small correction is also expected from the geometrical inductance of the 2-DEG, which can be calculated with the following relation²

$$L_G = 0.2l \left[\ln \frac{2l}{w+t} + 0.5 + 0.2235 * \frac{w+t}{l} \right] \text{ [nH]}$$

with l the length in mm, w the width and t the thickness of the 2-DEG. As $t \ll w$, this expression is independent of t . According to our geometry we have $L_G \approx 0.85 \text{ nH} \ll L_1$.

Taking into account both the tolerance factor on L_1 and the contribution of the geometrical inductance, we obtain the uncertainty on the determination of C_{sto} and a corresponding accuracy on J_s better than 15% in the entire phase diagram. This error margin is indicated by the grey outline on figure R3 below and on the Figure 4a of the main text.

Figure R3 : Experimental value of L_k extracted from the resonance frequency. The grey outline indicates the error margin.

The total inductance of the wire-bonds (1nH/mm per bond) can be neglected as we connected several bonds in parallel (>20 bonds) and as they are entirely covered by silver epoxy to further reduce their inductance.

Capacitance of the circuit. The parasitic capacitance of the sample circuit is difficult to calculate exactly as it involves different contributions (CPW line, contribution of MgO substrate, 2-DEG capacitance to ground...), but it can be determined experimentally. Figure R4 shows the temperature dependence of the resonance frequency of the sample circuit in the temperature range [0.4-300K]. It illustrates clearly the quantum paraelectric nature of STO. Between 300K and 10K, the resonance frequency $f_0 = 1/2\pi\sqrt{L_1 C_{sto}}$ increases by a factor ≈ 3.5 corresponding to an increase by a factor ≈ 12 in the total capacitance. The saturation of f_0 above 200K provides the value of the parasitic capacitance $C_{para} \approx 3.5 \text{ pF}$. Below 10K, quantum fluctuations lead to a saturation of the dielectric constant and f_0 is temperature independent. This is in particular true in the temperature range of interest (<500mK) as already seen in Fig

² Radio Engineers Handbook, McGraw-Hill, New York, 1945.

3a of the manuscript. Because of the gigantic value of the STO dielectric constant ($\epsilon_r \approx 25000$), the total capacitance of the circuit is mainly dominated by C_{sto} ($\approx 45\text{pF}$ at $V_G=0$). Although it is the total capacitance of circuit that is measured in the normal state ($C_{sto}+C_{para} \approx C_{sto}$) and plotted in Figure 2c, we refer to it as C_{sto} for sake of clarity. In any case, the presence of a small additional contribution due to the **parasitic capacitance (in parallel with C_{sto}) doesn't affect the determination of L_K** as the resonance frequency directly gives access to the total capacitance of the circuit for any gate voltage, which includes all the contributions.

Figure R4 : Resonance frequency of the sample circuit as a function of temperature.

Protective capacitors. As mentioned in the manuscript, the coupling capacitors C_p in series with R_1 and L_1 do not contribute to the RF response of the circuit in the frequency range of interest (0.2GHz-0.5GHz). Because of their large value ($2\mu\text{F}$) they behave essentially as shorts ($Z=1/C_p\omega \sim 1e-4$ ohms). This can be checked on the figure R5 below, which shows a numerical simulation of the reflection coefficient with and without the protective capacitors C_p .

Figure R5 : Simulated reflection coefficient Γ of the sample circuit with protective capacitors C_p and without.

Au/Ti strips. The sample is connected with multiple wire-bonds that pass through the 8 μm LAO layer to ensure an ohmic contact between the CPW line and the sample. Nevertheless, one can estimate the value of the "coupling capacitance" in the absence of wire-bonds by considering a plane capacitor geometry, where the two plates are respectively the 2-DEG and the Au/Ti gold strips separated by 8 μm thick LAO layer ($\epsilon_{\text{LAO}} \approx 20$). This gives a coupling capacitance of 17 nF (corresponding to an impedance of ≈ 0.02 ohms at f_0), which is much larger than C_{sto} (and therefore negligible as it is in series with the sample).

Conclusion :

Our resonant circuit has been extensively tested at low temperature using SMD devices, STO crystals and LAO/STO samples. With this kind of resonant experiment it is not possible to probe metallic superconductors such as Nb or Al as their low dc resistance in the normal state (compared to Z_0) will make the resonance invisible.

As discussed above, the main source of uncertainty in the determination of L_K are due to the contribution of the geometrical inductance and the tolerance on the value of L_1 . Taking into account these values the accuracy in the determination of L_K and correspondingly J_s is better than 15% in the entire phase diagram.

Changes : We have added a discussion on the possible sources of errors in our measurement method in the Supplementary Material part II. Figure R4 showing the temperature variation of C_{sto} has also been added. The issue of the geometrical inductance and the parasitic capacitance is now mentioned in the text and the information on the accuracy on the determination of J_s has been added (page 5 last paragraph) : "The total error, corresponding to the grey outline on figure 4a, is estimated to be lower than 15% for all gate voltages (see Supplementary Material)." The error margin on the determination of J_s has been added on Figure 4a and inset of Fig 4a.

Raw data and calibration

First, we would like to emphasize here that the resonance frequency in the normal state and at $T=20\text{mK}$ can be directly measured from uncalibrated raw data (S_{21}) as seen in the figure R6 below. This is one of **the main advantage of our resonant experiment**, which provides an extremely direct measurement of L_K . This important point was not mentioned in the first version of the manuscript. Of course the main drawback of this method is that the bandwidth is reduced and it is not possible to address the frequency dependence of $G(\omega)$ on a large frequency range (comparable with Δ).

Although raw data are already sufficient to obtain a quantitative measurement of L_K , the calibration allows suppressing parasitic signals mainly due to wave interferences in the microwave set-up. The precision on the measurement is also improved which is in particular interesting to study the temperature dependence or some gate voltage ranges corresponding to small shifts of f_0 .

Figure R6 : Comparison between S_{21} measured experimentally and the reflection coefficient Γ after the calibration procedure.

Changes: A sentence was added in the manuscript to specifically mention that L_K can also be directly extracted from raw S_{21} data (page 5, paragraph 1) : "The superconducting transition observed in dc resistance for positive gate voltages V_G , coincides with a shift of ω_0 towards high-frequency (Fig. 3b,c,d). We emphasize that this shift can already be detected in the uncalibrated $S_{21}(\omega)$ coefficient (Supplementary Figure SF4)."

A section was added on the calibration procedure in the Supplementary Material (Part III) including figure R6.

Calibration method

As explained in the manuscript a calibration is used to relate the measured quantity S_{21} to the reflection coefficient $\Gamma(\omega)$ of the sample circuit. In general, the calibration of any two-port linear microwave network requires the determination of three error complex coefficients³. This is in principle done using three known reference impedances, namely a short, an open and a match. Such calibration procedure is for instance available in standard Vector Network Analyzer. This procedure is not adapted to our experiment mainly for two reasons :

- The short, open and match standards have to be placed at low temperature (instead of the sample PC-board) and successively cooled down in the dilution refrigerator to 20mK with exactly the same experimental conditions. In practice this is extremely difficult as many parameters can slightly vary from one run to another (biasing conditions of the cryogenic amplifier, cooling power, thermalization of the cables and microwave components, tightening of all connectors...). **But most of all**, the standards are usually not qualified for cryogenic temperatures.
- For an optimal calibration, it is necessary to also include the last connector to the PC-board and the CPW line itself (otherwise the length of the circuit is different and wave interferences are not taken into account properly).

³ Pozar, D. M. *Microwave engineering 4th edition*, John Wiley & Sons (2012).

To solve this problem we have developed an (original) calibration method that employs the sample itself as a standard. This is possible in our experiment because (i) both the sheet resistance and C_{sto} can be varied with gate and (ii) their value can be reliably determined from the resonance (f_0 and magnitude of the absorption dip).

In the following we detail the calibration procedure that was used in the article. Although we consider a model that "mimic" the geometry of the experiment to make the method accessible to non-specialist readers, we would like to emphasize that the following results can be applied to any two-port linear microwave network through the signal flow graph theory³.

Figure R7 : Modeling of microwave set-up using the scattering matrix formalism.

The microwave set-up can be modeled using the scattering matrix formalism as shown in figure R7. a_i and b_i denote the complex amplitude of the incoming and outgoing waves at point i . $\alpha(\omega)$, $\beta(\omega)$, $\gamma(\omega)$ and $\delta(\omega)$ are complex coefficients representing the transmission and reflection coefficients in the circuit. They satisfy the following relations :

$$a_3 = \delta b_3 + \alpha a_1$$

$$b_2 = \gamma a_1 + \beta b_3$$

From these two equations, we obtain the relation between the transmission coefficient $S_{21}(\omega)=b_2(\omega)/a_1(\omega)$ between port 1 and port 2 which is measured with a Vector Network Analyzer, and the reflection coefficient of the sample circuit $\Gamma(\omega)=b_3(\omega)/a_3(\omega)$:

$$S_{21} = \gamma + \frac{\alpha' \Gamma}{1 - \delta \Gamma}$$

where $\alpha' = \alpha \beta$

A calibration procedure requires three known values of $\Gamma=(Z_L-Z_0)/(Z_L+Z_0)$ to determine the coefficients α' , γ and δ . In our case, three different values of the gate voltage allow realizing three different impedances Z_L whose value can be reliably determined from the resonance (f_0 and magnitude of the absorption dip). As we can tune the gate continuously, we can realized a large number of different impedances and check that the final calibration is independent of the chosen gate values.

Changes : A section devoted to the calibration method, including figures R6 and R7, has been added to the Supplementary Material.

Temperature dependence of the superfluid stiffness

As explained in the previous paragraphs, our resonant experiment provides a very direct measurement of the superfluid stiffness J_s independently of the calibration. A quantitative agreement with the BCS gap (Eq. (2)) is obtained in the overdoped regime without any adjustment parameters by combining RF (J_s) and dc (Rn) measurement. This confirms the validity of the method to measure the absolute value of the superfluid stiffness.

As mentioned by the referee, the temperature dependence of the resonance frequency shift could also be used to obtain some information on the superconducting 2-DEG. More specifically, a simple calculation shows that

$$\frac{\omega^2(T) - \omega_0^2}{\omega_0^2} = \frac{L_1}{L_k} = L_1 \frac{4e^2}{\hbar^2} J_s(T)$$

where ω_0 is the resonance frequency in the normal state and $\omega(T)$ is the temperature-dependent resonance frequency in the superconducting state. The temperature dependence of

the superfluid stiffness is given by the following expression $J_s(T) = J_s(0) \frac{\Delta(T)}{\Delta(0)} \tanh\left(\frac{\Delta(T)}{2k_B T}\right)$

which is an interpolation between two temperature regimes⁴. A fit of $\frac{\omega^2(T) - \omega_0^2}{\omega_0^2}$ will

therefore provide an estimate of the gap energy as this latter appears in an un-normalized form into the hyperbolic tangent term. However, it is not possible to access to $J_s(0)$ without knowing the value of L_1 which is a crucial parameter in our analysis as described previously.

Nevertheless, analyzing the temperature dependence of $J_s(T)$ is still relevant as it allows extracting the gap energy $\Delta(0)$ which can then be compared with the value converted from $J_s(0)$ through equation (2). In figure R8 we show an example of a fit of the low temperature part of the normalized stiffness $J_s(T)/J_s(0)$ using the expression given in inset. The value of Δ_0 obtained from the fit is 23.4 μeV which confirms the value given by equation (2) (ie $\Delta_0=22.2$ μeV). As shown in figure 4b this value is very close to $1.76k_B T_c$ for this gate voltage corresponding to the overdoped regime.

As seen in figure R8, the temperature dependence of $J_s(T)$ reveals a jump which raises the important issue of Kosterlitz Thouless physics in superconducting interfaces. This issue is addressed in the response of referee 2 on page 23 of this response letter. In this article we decided to discuss essentially the phase diagram on the basis of the $T=0$ quantities. These general results can be understood without making reference to any specific theory, leaving for future work a detailed analysis of the temperature dependence.

⁴ Alternatively, one can use some phenomenological models as in ref 10.

Figure R8 : Temperature dependence of the normalized stiffness at $V_G=34V$ fitted using the theoretical expression in inset. The fit gives $\Delta(0)=23.4 \mu eV$.

Referee's comment b) : " Since the Authors uses the Sheet resistance in the normal state measured in dc, it would be important to know what is the reason why they expect that this quantity is not frequency dependent. At the microwaves, the losses are determined by the microwave surface resistance, which could coincides with the sheet resistance only if the are not additional scattering terms (due to disorder for example) which contribute."

Sheet resistance

Experiments are performed at frequencies (<600 MHz), which are much lower than the typical scattering rates. For this reason we expect the sheet resistance to be frequency independent in the frequency range of interest. The amplitude of the absorption dip, which is controlled by the total dissipation of the circuit follows the variation expected from the gate dependence of R.

As already mentioned in the previous paragraphs, the determination of the resonance frequency f_0 (and $G_2(\omega)$) doesn't dependent on the resistance of the circuit as long as it is sufficiently close to $Z_0=50$ ohms to generate a clear absorption dip.

Changes : we made this point clearer in the Supplementary Material part II : " The determination of the superfluid stiffness J_s relies on the measurement of the kinetic inductance L_K , which is extracted from the resonance frequency ω_0 in the superconducting state. This latter depends only the total inductance and capacitance of the sample circuit. The total resistance of the circuit is not involved in the determination of J_s but it must be sufficiently close to $Z_0=50$ ohms to generate a visible absorption dip."

c) Referee's comment : "Is the T_c used in the paper to calculate the BCS gap determined from the Resistive dc measurement? Is it the zero resistance T_c ? Are these measurements performed on the same device with the microwaves applied to the sample or not?"

Our answer : T_c is defined as the temperature where the dc sheet resistance R reaches zero. This information was given in the caption of figure 4 but was missing in the text. dc measurements are performed at the same time than RF measurements thanks to the bias-tee (Cp capacitors avoid the dc current to flow in R_1 and L_1). The typical ac current flowing into the sample circuit, which is imposed by the microwave power delivered by the VNA, is less than 5 nA. This is much lower than the critical current of the superconducting 2-DEG ($\approx 5 \mu\text{A}$). We also varied the RF power to check that it doesn't affect the dc measurement (including turning OFF the RF)

Changes : We have added the definition of T_c directly in the text (page 5 second paragraph) : " In Figure 4a, we plot the gate dependence of the experimental superfluid stiffness

$$J_s^{\text{exp}} = \frac{\hbar^2}{4e^2} L_k \text{ extracted from } L_k \text{ at the lowest temperature } T = 20 \text{ mK } (\approx 0 \text{ K in the following}).$$

On the same logarithmic scale, we also show the gate dependence of the superconducting T_c defined as the temperature where $R_{dc} = 0 \Omega$."

Information on the RF power in the manuscript (page 4 first paragraph) : "The typical ac current flowing into the sample circuit, which is imposed by the microwave power delivered by the VNA, is less than 5 nA, which is much lower than the critical current of the superconducting 2-DEG ($\approx 5 \mu\text{A}$)."

The work presented in the manuscript results from a long experimental development. A careful attention has been devoted to the design of the microwave circuit to minimize all possible sources of errors. Many test experiments have been performed to validate the method. For all these reasons, we are confident that the data presented here are entirely reliable.

I. Remarks about the main messages

d) referee's comment : *A loss of phase coherence is only one of the possible explanation for a reduction of the superfluid stiffness respect the BCS value in a superconductor. It is well known that the penetration depth measured at the microwaves can vary a lot even in standard superconductors due to extrinsic contribution, related to defects, like point, linear or other kind of defects. Thus the Author should explain why other scenarios are not plausible to explain the data, or smooth out their conclusion about this part of the paper, which is not, in my opinion, the most original and important one.*

Our answer : We completely agree with the Referee, and indeed our analysis already accounted for the extrinsic defects. Eq. (2) provides the stiffness of the superconductor based on the dirty BCS limit. Therefore it already includes the extrinsic mechanisms that the Referee correctly mentions through the normal resistance. While Eq. (2) works well in the overdoped regime, in the underdoped one we find a strong suppression of J_s^{exp} with respect to the BCS value.

The Referee's question probably is due to a misunderstanding related to our sentence in page 6 "This indicates that the global phase coherence of the superconducting condensate is partially lost in the 2-DEG". This sentence is indeed ambiguous and we apologize for that. The superfluid stiffness is by definition a measure of the phase rigidity of a superconductor. Thus any mechanism suppressing such a phase coherence automatically is mirrored in a decrease of J_s .

Changes : To make this point clearer, we revised the manuscript in two points:

1- we added a brief sentence after eq. (2) to clarify that it accounts for the scattering by defects.

2- we revised the sentence in page 6 as : This indicates that the loss of phase coherence of the superconducting condensate is stronger than what expected taking into account conventional scattering by defects, as encoded in Eq. (2). Such a behaviour can then be ascribed to strong phase fluctuations....

Referee's comment *The idea that in the underdoped phase LAO/STO is essentially composed by a network of 2D-SC islands Josephson coupled has been already anticipated by the Authors in previous publications. In the LAO/STO case, however, there are several studies on nano bridges [Appl. Phys. Lett. 101, 222601 (2012), Nano Lett. 15, 2627 (2015).], suggesting that the LAO/STO 2DEG is quite homogeneous at the nanoscales. How the Authors explain this contradiction with other data present in literature?*

Our answer :

In reference [11] we proposed to model the 2-DEG by an array of superconducting island coupled by Josephson effect. Depending on the temperature and the region of the phase diagram, the characteristics of the islands can be very different. In the overdoped regime, the islands are robust and tightly connected at $T = 0$ (homogeneous-like), whereas in the UD regime, the array is more dilute. Signatures of Josephson physics, in particular switching of critical current, have also been reported on LAO/STO samples (with $T_c > 300\text{mK}$) in reference [30]. There are nowadays many other experimental reports that confirm the presence of disorder in oxides interfaces at various scales in LAO/STO interfaces [see for instance Ariando et al., Nat. Commun. 2, 188 (2011), B. Kalisky et al., Nat. Mater. 12, 1091 (2013). M. Honing et al. Nat. Mater. 12, 1112 (2013), Frenkel et al. ACS Appl Mater Interfaces.8, 12514–12519 (2016), Feng Bi et al. Journal of Applied Physics **119**, 025309 (2016), Caprara et al Phys. Rev. B 88, 020504(R) (2013).] However, this is not in contradiction with the works cited by the referee as the system can still be rather clean in some regions of the phase diagram and at a sufficiently small scale. In particular in the overdoped regime, we recover the homogenous like properties at $T \approx 0$. For instance, in reference [11] we extract an order of magnitude of ~ 100 nm for the typical size of the islands (~ 200 nm in [31]). This scale is comparable to the size of the nanobridges used in the mentioned references. More over, in these works the properties of the nanobridges are characterized by dc transport, which vary rather smoothly over the phase diagram. The stiffness instead is a very sensitive probe of inhomogeneities since it is rapidly suppressed by any dephasing mechanism.

Referee's comment : *"Additionally the non-linear dielectric constant of STO, usually by itself give rise to a shift of the resonant frequency (also measured by the Authors) which can be sample dependent."*

Our answer :

Indeed, the non-linear dielectric constant of STO produces a shift of the resonance frequency with gate voltage (as seen on Figure 2). C_{sto} can not be calibrated once and for all because the dielectric constant varies from sample to sample and even from one run to another when cycling the sample at 300K. This is why C_{sto} is systematically measured (through the resonance frequency) for all gate voltages as a function temperature during the experiment. In the temperature range of interest, C_{sto} has no temperature dependence. This is confirmed by

the data of Figure 3a where we see that f_0 is constant in the entire temperature range when the 2-DEG is not superconducting ($V_G = -34V$). f_0 is also constant above T_c for all gate values. The shift of the resonance frequency below T_c can therefore be attributed to the kinetic inductance of the superconducting 2-DEG.

Changes : Information on the temperature dependence of C_{sto} has been added in the text (page 5 first paragraph) : "In absence of superconductivity (for $V_G < 0$ V), the resonance frequency remains unchanged as C_{STO} as no temperature dependence in the range of interest (Fig. 3a)." This issue on C_{sto} is further discussed in the Supplementary Material (part II and supplementary figure SF2)."

e) Referee's comment : *The gap and the superfluid density are estimated by making the assumption that LAO/STO is dirty single-gap BCS superconductor. Some of these assumptions are not fully justified.*

Our answer :

Single gap : In this article we consider the most simple case of single band superconductivity in agreement with tunneling experiments (ref 9) and SQUID magnetometry measurements (ref 10). A more recent work on tunneling experiments (*Phys. Rev. B* 96, 014513 (2017)) revealed the presence of in-gap states in some LAO/STO samples. The possibility of a two-gap superconductivity is mentioned by the authors before they conclude that multiband superconductivity is finally unlikely. Recently, Josephson experiments (Stornaiuolo et al, ref 32) suggested that a second unconventional gap could also take place due to the interaction between superconductivity and spin-orbit coupling.

Our experiment measures the superfluid stiffness J_s of the 2-DEG and the gap energy is only obtained by converting J_s through equation (2). In general, two-band superconductivity can only be evidenced in superfluid stiffness measurements if the contribution of each band is comparable. Consequently a band with a very low superfluid density will therefore be difficult to see in presence of an other band with a higher superfluid density. For this reason our experiment is probably not very well adapted to probe the existence of two gaps⁵. As discussed in the response to the next comment, the gap that we derived here is very similar to the BCS one reported in ref 32 (not the unconventional one).

Dirty-limit : We agree that the justification of the dirty limit was missing. The dirty limit for a superconductor is characterized by an elastic scattering time τ much shorter than the gap energy Δ_0 , ie $\frac{\Delta\tau}{\hbar} \ll 1$ (or equivalently $l \ll \xi$). In our case, $\tau \approx 1.5e-13$ s (for $V_G = 50V$) and the

gap energy is around 23 μ eV. This gives a ratio $\frac{\Delta\tau}{\hbar} \approx 0.0055 \ll 1$, which indicates that the system is in the dirty limit.

Changes : information on the dirty limit was added (page 5, last paragraph) : "The superconducting 2-DEG is in the dirty limit in which the elastic scattering time τ is much

⁵ Note that in some peculiar situations (comparable superfluid densities), multiband superconductivity can be identified in the temperature dependence of the superfluid stiffness.

shorter than the superconducting gap $\Delta(0)$, $\frac{\Delta\tau}{\hbar} \approx 0.0055 \ll 1$. Within this limit and for $\omega \ll \Delta(0)/\hbar$, the zero-temperature superfluid stiffness of a single-band BCS superconductor can be expressed as a function of $\Delta(0)$..."

Referee's comment : *Recent works show that SC in LAO/STO could be rather unconventional [Phys. Rev. B 95, 140502 (2017), Phys. Rev. B 96, 014513 (2017)]. In my opinion, the determination of the gap from the superfluid stiffness is only an indirect indication of a single, BCS, gap. In general, the present study is sensitive to the main pairing channel, i.e. the pairing channel which carries most of the superfluid density contribution. If another channel is present, maybe just in some fraction, and it is unconventional, it is clear that it could be much more sensitive to the experimental conditions/methods. For example microwave radiation could destroy any contribution to σ_2 of an unconventional order parameter, while being able to measure the major contribution to the superfluid stiffness due to the conventional channel.*

Our answer : We are aware of these recent and very exciting findings. As mentioned in the previous response, our experiment was not designed to address this issue. As far as our work is concerned, we haven't observed any feature that would suggest the presence of an additional unconventional phase. However, that was not the focus of our work and of course it doesn't mean that this unconventional phase does not exist. From the experimental point of view, we carefully checked that the determination of L_K doesn't not depend on the microwave power (down to the lowest measurable signal).

Changes : We have added an information on the microwave power in the text (page 4 first paragraph) : "The typical ac current flowing into the sample circuit, which is imposed by the microwave power delivered by the VNA, is less than 5 nA, which is much lower than the critical current of the superconducting 2-DEG ($\approx 5\mu\text{A}$)."

f) Referee's comment : *The gap obtained at optimal doping in the present work is in better agreement with spectroscopic data in ref. [Phys. Rev. B 95, 140502 (2017)] (for one of the two gaps found, the BCS one), but is half of the value reported by Mannhart et al. This discrepancy should be discussed.*

Our answer : We thank the referee for pointing out that our measurement is in agreement with the BCS gap measured by Stornaiuolo *et al.* in spectroscopic Josephson junctions made on sample with similar T_c . We find the same BCS agreement than Mannhart *et al.* ($1.7K_B T_c$) at optimal doping but our gap is smaller in agreement with our reduced T_c . Literature on LAO/STO reveals an important spread on T_c at optimal doping, typically in the range [100mK-300mK] which is not well understood. Our T_c is similar to the one found in PRL 107, 056802 (2011) for instance or in Phys. Rev. B 95, 140502 (2017). T_c could depend on sample growth conditions, carrier density or intrinsic disorder. In the present case, we have a sample with a rather high carrier density (close to the density predicted by the polar catastrophe scenario), which very often tends to give a lower T_c . We have used the same measurement technique on samples with a higher T_c ($\approx 220\text{mK}$) and found a larger gap ($\approx 34\mu\text{eV}$) with a similar BCS agreement.

Changes : we have added the reference Phys. Rev. B 95, 140502 (2017) [ref 32] and a sentence regarding the values of the gap and of T_c compared with the ones reported by Mannhart *et al.*(page8) : This is in agreement with the BCS gap identified recently by Stornaiuolo *et al* at optimal doping using spectroscopic Josephson junctions in LAO\STO interfaces of similar T_c [32]. By using tunneling spectroscopy on planar Au\LAO\STO junctions, Richter *et al.* have reported an energy gap in the density of states of $\approx 40\mu\text{eV}$ for optimally doped LAO\STO interfaces of higher T_c [9]. In spite of this significantly higher gap energy, this corresponds to a $\Delta/k_B T_c$ ratio of 1.7, similar to our result".

e) referee's comment : *Fig. 5d and Fig. 5e reports the density of $3d_{xy}$ and $3d_{xz,yz}$ carriers and the superfluid density estimated from the microwave measurements. There is very likely an error in the vertical scale (a factor 10?) of Fig. 5d.*

Our answer : There is no error in the vertical axis of Figure 5d. The total carrier density is $n=2.10^{14}$ e/cm² at maximum doping close to the density predicted by the polar catastrophe scenario. As mentioned in the text, the superfluid density represents approximately 1% of the total carrier density.

g) Referee's comment : *Still in Fig. 5e the superfluid density and the density of high mobility carriers are of the same order of magnitude but are different. In particular, it seems that there are other carriers contributing to superfluid density. Which carriers? Is moreover correct to assume a constant effective mass in the calculation of the superfluid density? Some care is needed about that.*

Our answer : The comparison of n^{2D} s with n_{HM} shows that both quantities have a very similar gate dependence and agree "semi-quantitatively". Although n_{HM} and n^{2D} s are obtained through rather direct experimental methods (Hall effect and resonant microwave transport), their exact determination still requires some assumptions. For the Hall effect we assume that only two bands are involved whereas several d_{xy} subbands are occupied in principle. This could modify the determination of n_{HM} . In the resonant transport experiment we measured the superfluid stiffness and not the superfluid density. As mentioned by the referee, we assume a constant mass to relate these two quantities but the real picture could be more complicated, for instance in presence of spin-orbit coupling (atomic and Rashba spin-orbit coupling) that would modify the band structure. In this article, we relate n_{HM} to the density of electrons in the $d_{xz/yz}$ band and propose that the superconducting condensate is mainly formed by these electrons. However, we can not exclude that the presence of an interband coupling between the d_{xy} and the $d_{xz/yz}$ bands could also induce the condensation of a small number of electrons in the d_{xy} band.

In absence of a clearly established band structure for STO-based interfaces (which will in addition vary with the confinement conditions in each sample), it is difficult to go beyond the approach presented in this manuscript.

Changes : A sentence was added about the possibility to induce superconductivity in the d_{xy} band (page 9, last paragraph). "Nevertheless, in the presence of interband coupling, superconductivity may also be induced in some d_{xy} subbands which would then slightly contribute to the total superfluid density."

h) Referee's comment : *Why the T_c seems different in the dc-resistance measurement and in the microwave measurement?*

Our answer : The T_c defined by $R(T_c)=0$ in dc measurement corresponds to the temperature below which $J_s(T)$ starts to increase. As seen in figure R9, the two T_c are very close. However, the presence of the superfluid jump (see response to points (a) and (i)) may give the impression that the T_c in J_s is slightly shifted towards the low temperatures.

Changes : The definition of T_c as the temperature where $R_{dc}=0$ appears more clearly in the manuscript now (page 5 second paragraph) : " On the same logarithmic scale, we also show the gate dependence of the superconducting T_c defined as the temperature where $R_{dc}= 0$ Ohms."

i) **referee's comment** : *The temperature dependence of the superfluid stiffness is not discussed and not fitted by any model. Why? Additionally I notice some non monotonous behavior close to T_c in Fig. 3.*

Our answer : There is indeed some non-monotonous behavior close to T_c which may be related to BKT physics as mentioned in the response to point a). This issue is also addressed in the response to the second referee (page 23).

Response to referee B

We thank the referee for his/her critical reading of our manuscript, suggestions of improvement and support for publication. We detail below our responses to his/her comments, together with the changes we made on the revised version of the article. We hope that the new version of the manuscript will be suitable for publication.

referee's comment : *On the measurements of the superfluid density, the work of K. Moler is cited as reference 7 but does not appear anywhere in the text. Reference 10 - Bert et al. PRB 86 060503(R) (2012) must be discussed in some details since these authors measure precisely the superfluid density versus gate as in this paper. The authors should then stress what is different and what is new in their contribution?*

Our answer : We thank the referee for pointing out this problem with reference 7 which was deleted as the main topic of this article is the magnetism that is not addressed in our manuscript.

In the article by Bert et al. [PRB 86 060503(R) (2012)], the superfluid density J_s was measured using scanning SQUID magnetometry. In this technique, screening currents in response to a magnetic field are measured locally and converted into a magnetic penetration length (or equivalently the Pearl length for 2D superconductors), which is related to the superfluid stiffness. This is a very powerful scanning technique that offers a spatial resolution of a few micrometers (Bert et al, Nature Mat).

Our resonant method can only probe the macroscopic stiffness of the LAO/STO interfaces but it is comparatively simpler and more direct as it requires only one external parameter to determine J_s (the circuit inductance L_1). This is a crucial issue to obtain the quantitative agreement with the BCS theory that we report in the overdoped regime. From the experimental point of view the second difference is that in our experiment we can perform

both dc and RF measurements at the same time. We can therefore measure the three quantities J_s , R_n and T_c (resistive transition) independently as a function of gate voltage and **compare the evolution of J_s and Δ with BCS predictions through Eq. (2).**

We also measured the Hall effect and gate capacitance in the entire phase diagram, which allows us **comparing the superfluid density to the carrier density of the dxz/yz band.** These two aspects are precisely related to the important messages that we convey in this article.

Bert et al. determined the gap energy from the temperature dependence of the magnetic penetration depth through a phenomenological model. This technique can only give an estimate of the $T=0$ gap. They observed strong deviations from the BCS model not only in the zero-temperature gap ($\Delta=2.2k_B T_c$ instead of $1.76 k_B T_c$) but also in the temperature dependence (exponent $a=1.4$ instead of $a=1$ for BCS). Moreover, they didn't observe any evolution in the phase diagram. These results are in contradiction with the gap energy $\Delta(T)$ reported by Richter et al. in tunneling experiments [Nature 502, 528-531 (2013)] and by Stornaiuolo et al. in spectroscopic Josephson junctions (BCS gap) [Phys. Rev. B 95, 140502 (2017)]. This is also different with the results reported in our manuscript for the $T=0$ gap energy and also for the temperature dependence (see figure R9 at the end of this response letter).

As far as the superfluid density is concerned, its evolution with gate voltage reported by Bert et al. is similar to our. By combining RF and dc transport we were able to show (in addition) that while the superfluid stiffness increases continuously with gate, the gap energy (converted from the stiffness) follows the characteristic dome shape of T_c . The decrease of T_c in the overdoped regime is therefore not due to a loss of superfluid density. The maximum value of the superfluid density given by Bert et al. is $3 \cdot 10^{12} / \text{cm}^2$ is similar to the one we obtain ($2 \cdot 10^{12} / \text{cm}^2$) although they use a different hypothesis on the mass of carrier ($m=1.46m_0$ for the d_{xy} band). However, given the large uncertainty on their determination of the absolute value of n_s , this could be compatible with our result. We also note that the carrier density is probably higher in our case ($\sim 10^{14} / \text{cm}^2$) than in their sample ($\approx 2 \cdot 10^{13} / \text{cm}^2$ measured in a different cool down with no gate).

Changes : Several sentences were added in the text to stress the specificity of our experiment of put forth our contribution.

We stressed that our measurement allows measuring both the dc and RF properties at the same time and that the determination of superfluid stiffness is very direct (page 4) : "A reflection measurement gives therefore a direct access to the load impedance $Z_L(\omega)$ or equivalently its admittance $G_L=1/Z_L$, commonly called complex conductance. In the present case, Z_L is the impedance of the RLC circuit represented in Fig. 1c., whose resonance frequency ω_0 in the superconducting state is directly related to kinetic inductance of the 2-DEG. Measuring ω_0 as a function of gate voltage, provides therefore a very direct method to determine the superfluid stiffness in the phase diagram. In addition, the set-up of Figure 1, which includes a bias-tee and protective capacitors in series with L_1 and R_1 allows measuring both the dc and ac microwave transport properties of the 2-DEG at the same time."

We also specifies the accuracy of the measurement : "The total error, corresponding to the grey outline on figure 4a, is estimated to be lower than 15% for all gate voltages (see Supplementary Material)."

In addition, a sentence was added on the comparison of the superfluid density (page 10, first paragraph). "Bert et al. measured the superfluid density in LAO/STO interfaces using a

scanning SQUID techniques [10]. The overall gate dependence is similar in both experiments, including in the OD regime where the superfluid density keeps increasing while T_c is reduced. However, in our case n_s is lower despite a much higher carrier density."

Referee's comment : *On the question of the orbital ordering, reference 12 should be the paper of Salluzzo and co-workers (PRL 102, 166804 (2209)) since it is the first experimental report on the orbital reconstruction. The authors may want to cite Berner et al. too.*

Our answer : We agree with the referee and we have added this reference.

Changes : The reference to the work of Salluzzo and co-workers was added (Ref [12])

Referee's comment : *On the high mobility / low mobility carriers, several papers should be cited: Bell et al. PRL 103, 226802 (2009) Ben Shalom et al. PRL 105, 206401 (2010) Fete et al. PRB 86, 201105(R) (2012) Joshua et al. (ref. 14) Nat. Comm. (2012)*

We agree that these are important contributions to the field.

Changes : References were added.

Referee's comment : *References 36 and 37 are not on the LaAlO₃/SrTiO₃ system.*

Our answer : These references are about LaTiO₃/SrTiO₃ which is very similar to the LaAlO₃/SrTiO₃ from the point of view of the superconducting 2-DEG. We have added this information in the text to avoid confusion.

Changes : We explicitly mention that multiband transport has been observed both in LAO/STO and LTO/STO interfaces (page 8, second paragraph) : "Multiband transport in LAO\STO and LTO\STO interfaces has been observed experimentally in various magneto-transport experiments including quantum oscillations [37,38] magneto-conductance [15,39]] and Hall effect [2,3,40,41]."

Referee's comment : *On the role of the dxz, dyz bands on superconductivity, Gariglio et al. (APL Mater. 4, 060701 (2016)) discuss in detail the superconducting phase diagram of the system, the key role of the heavy high-density of states dxz, dyz bands for superconductivity and the fact that this could explain the low superfluid density observed by Bert et al. On the theory side, Valentinis et al. (ArXiv 1611.07763) propose a scenario that describes the phase diagram of the LaAlO₃/SrTiO₃ system and explains the role of the heavy subbands.*

Our answer : The article by Gariglio et al. (APL Mater. 4, 060701 (2016)) was already cited and we emphasized more this contribution in the last part of the new version. The contribution of Valentinis et al regarding the heavy bands has been added.

Changes : the contribution of Gariglio et al. [6] and Valentinis et al [7] have been emphasized in the new version of the paper (page 2 second paragraph) : "This highlights the important role of orbitals ordering and also suggests that only some specific bands could host superconductivity [15]. In particular, it has been emphasized that the dxz/dyz band lying at

high energy in the quantum well could play an important role because of its large density of states [6,7].

Referee's comment : -p.2 - *The authors say that they propose that the filling of the dxz, dyz bands controls the emergence of superconductivity. As mentioned just above, this scenario has been proposed before by several authors. The text should thus be rephrased highlighting the fact that, in this paper, the carrier density in the dxz, dyz bands is compared to the superfluid density making this statement much stronger.*

Our answer : We agree that the possible connection between the filling of the heavy band and superconductivity has already been proposed. The sentence in the abstract was misleading and we rephrased it to stress our findings.

Changes : We modified the last part of the second paragraph in page 2 as : " Here we use resonant microwave transport to measure the complex conductivity of the superconducting (001)-oriented LAO/STO interfaces. This allows us to directly extract the evolution of the superfluid stiffness in the phase diagram that we also convert into a gap energy through BCS theory in the dirty limit. Both energy scales are compared with theoretical predictions. The superfluid density n_s deduced from J_s is found to be close to the carrier density of the dxz/dyz band extracted from multiband Hall effect measurements, highlighting the key role of this band in the emergence of superconductivity.

Referee's comment : -p.2 - *The authors say that they extract the superfluid stiffness and the superconducting gap from their measurements. As I understand, they determine the kinetic inductance of the superconductor as a function of the doping and from this calculate (in a BCS model) the gap – they do not measure the gap independently as one would do in a tunneling experiment. This point should be clarified.*

Our answer : Indeed, in this experiment we measured the kinetic inductance (or equivalently $G_2(\omega)$) which is directly related to the superfluid stiffness. The stiffness is then converted into a gap energy using the normal state resistance through BCS theory.

Changes : We modified the last part of the second paragraph in page 2 to clarify this point. " This allows us to directly extract the evolution of the superfluid stiffness in the phase diagram that we also convert into a gap energy through BCS theory in the dirty limit. Both energy scales are compared with theoretical predictions..."

In the first paragraph in page 8 we also explain that our experiment probe the phase coherence and not the gap in the density of states as in a tunneling experiment. "In our case, the superconducting gap Δ_s^{exp} probed by microwaves is directly converted from the stiffness of the superconducting condensate and is therefore only reflective of the presence of a true phase-coherent state. On the other hand, tunneling experiments probe the single particle density of states, and can evidence pairing even without phase coherence."

referee's comment : *The technique used to determine L_k is very interesting. Figure 1 however does not allow the reader to understand the experimental set-up and the measurements. Improving the sketch of the device would be useful as well as a better description of the measurements themselves.*

Our answer :

We agree with the referee that information on the measurement was missing in the first version of the manuscript.

Changes : We have included a large section devoted to this issue in the Supplementary Material of the revised version. It includes information on the sample, pc-board, microwave set-up and calibration.

Referee's comment : *From Figure 2.c, one can probably extract the dielectric constant of SrTiO₃ – it would be interesting to plot the latter as a function of the gate.*

Our answer : The referee is right. As seen in figure R4 of this response letter (and figure SF2b in the supplementary material), the capacitance of the circuit is entirely dominated by the STO substrate and the data of figure 2c can be used to extract the STO dielectric constant ϵ_r . However, it is not straightforward as there is no analytical expression relating the two quantities, which is valid in our exact geometry. We have performed numerical simulation using finite element method software (Comsol) to calculate the dielectric constant (at $\omega = 0$) corresponding to the measured capacitance. We assume that the LAO layer doesn't contribute (too thin) and we consider a STO substrate (nominally 0.2 mm thick as checked carefully and not 0.1 mm as previously indicated) with 0.1 mm wide Au/Ti strips put on each side of the substrate. A one volt potential is imposed on one Au/Ti strips while the other one is at the ground (zero volt). Figure R9a shows the distribution of the potential (color scale) in the STO and the direction of the electric field (arrows). According to the simulation, the corresponding capacitance between the Au/Ti strips is directly proportional to the dielectric constant (and roughly proportional to the cross sectional area). Figure R9b shows the gate dependence of ϵ_r obtained from the simulation which corresponds to the measured STO capacitance (parasitic capacitance has been subtracted). $\epsilon_r \approx 23700$ for $V_G = 0V$ which is consistent with values reported in the literature. The total error (grey outline) is mainly given by the uncertainty on the exact thickness of the STO substrate, which has been found to be $\pm 10 \mu m$.

Changes : We have added a sentence about the dielectric constant of STO in the text (page 5, paragraph 1) : " According to the geometry of the sample, its value at $V_G = 0V$ ($\approx 42 pF$) corresponds to a dielectric dielectric constant $\epsilon_r \approx 23700$ (see Supplementary Figure SF4)". We have also added a paragraph on the dielectric constant in the Supplementary Material including figure R10.

Figure R9 : a) Numerical simulation of the STO capacitance using finite element method. The color scale corresponds to the value of the electrostatic potential in the STO and the arrows indicate the direction of the electric field. b) STO dielectric constant extracted from the simulation as a function of gate voltage. The grey outline indicates the error margin.

referee's comment : *An important comment concerns Fig. 4 and the discussion of the data that I find confusing. The equations that are used: the BCS gap calculated from T_c and the relation between the superfluid stiffness and the gap are valid for a BCS superconductor (for a mean field transition). If one is having a Kosterlitz-Thouless like transition – a transition controlled by phase fluctuations, the establishment of phase coherence is not linked to the gap and one thus does not get the gap from the phase stiffness – the two equations are thus valid in the overdoped regime but not in the underdoped one.*

Now, I think I understand what the authors want to do; they want to demonstrate that the underdoped part of the phase diagram is not BCS-like. Fine, but the way the data are presented is highly confusing – Fig. 4 shows Δ_{BCS} versus gate – at the QCP, Δ_{BCS} obviously goes to zero since (the establishment of phase coherence) T_c goes to zero (as well as Δ_{exp} – making a comparison with the tunneling data (p.6) irrelevant). At the QCP, however, the mean field T_c and the pairing gap may well be finite – very large if one is believing the Stuttgart tunneling data – whatever the behavior of the gap is, the Figure is very misleading for the reader since one is seeing Δ_{BCS} versus gate as the equation used is not valid in the underdoped part of the phase diagram.

Our answer :

We agree with the Referee that Eq. (2) is indeed valid only for dirty BCS superconductors. In general, in the presence of quantum or thermal phase-fluctuation effects Eq. (2) only provides an upper bound for the real stiffness, which is in general suppressed by phase fluctuations. This is indeed shown in the upper panel of Fig. 4, where the experimental stiffness is compared to the estimated one based on Eq. (2). On the overdoped regime, a good agreement is obtained with BCS theory whereas in the underdoped regime, the strong suppression of J_s^{exp} indicates that the loss of phase coherence of the superconducting condensate is stronger than what is expected from a simple BCS picture (J^{BCS}).

In the lower panel of Fig. 4 we wanted to convey the same message of the upper panel, but converting now the stiffness in the gap energy scale. Moreover, this shows that while the stiffness increases continuously in the overdoped regime, the gap energy obtained from Eq.

(2) reproduces the characteristic dome shape behavior of T_c . To be more precise, the red curve is nothing else than the BCS scale $1.76k_B T_c$, and as such it must vanish when T_c vanishes. The blue curve is instead the gap extracted from Eq. (2): the fact that it is lower than the red one in the underdoped regime just reflects the additional suppression of J_s with respect to the expected BCS value. As far as the comparison with the tunnelling gap is concerned, all the discussion in Page 6 has the goal to stress exactly the point raised by the Referee: our RF measurement probes phase coherence and not directly pairing, which could very well survive above T_c .

Changes : To avoid any confusion we removed the misleading quantity Δ_{BCS} in the text and in the figure where it was replaced by $1.76k_B T_c$. In page 6, we added a sentence regarding the suppression of J_s^{exp} in the underdoped regime, "This indicates that the loss of phase coherence of the superconducting condensate is stronger than what expected taking into account conventional scattering by defects, as encoded in Eq. (2)."

In page 7, we explain that our method doesn't probe the pairing gap in the single particle density of state as in a tunneling experiment. We also added a sentence to explicitly indicate that the pairing gap can not be extracted from Eq. (2) in the underdoped regime as the transition is dominated by the loss of phase coherence. "In this case, the pairing gap can not be extracted from Eq. (2) which is valid only for BCS superconductor."

Referee's comment : *What about a Kosterlitz-Thouless transition? This point is essentially not discussed as the authors mention phase fluctuations and the fact that the system behaves as a Josephson junction array whose physics is the one of a 2D xy model. This issue should be discussed in detail keeping in mind that, although disorder and/or inhomogeneities in the superfluid density may wash out the signatures of a BKT transition, such a transition may take place.*

Our answer : We agree with the Referee that transverse (vortex-like) BKT effects could be relevant in superconducting interfaces. However, BKT physics is relevant near T_c , where transverse vortex fluctuations can be thermally excited, not in the $T=0$ limit that is the focus of our manuscript. More specifically, the suppression of the stiffness with respect to the BCS estimate Eq. (2) that we observe in the under doped regime can be due to (longitudinal) quantum phase fluctuations, and do not have to be linked necessarily to BKT physics. In particular inhomogeneity can enhance the quantum phase-fluctuations corrections to the stiffness at $T=0$, as shown by recent theoretical work by different groups [G. Seibold et al. Phys. Rev. Lett. 108, 207004 (2012), Mason Swanson et al, PRX 4, 021007 (2014)].

For what concern the possibility that BKT physics is present in interfaces, the issue itself is still debated. Indeed, as the Referee correctly mentions, after the first proposal by the Geneva group (ref 1) that BKT physics could be present, several works pointed out that the typical BKT effects could be smeared out by the presence of inhomogeneity. Our data confirm this interpretation, as one can see in Fig. R10 where we show the temperature dependence of $J_s(T)$ for an optimally-doped sample. Even though the low-temperature part can be very well reproduced by a conventional BCS fit, the data deviates from it in a BKT-like jump. However, the jump occurs before the intersection with the universal line, and it is smeared out. Similar effects have been seen also in conventional superconducting thin films, and attributed to low vortex-core energy [M. Mondal et al., Phys. Rev. Lett. 107, 217003 (2011)] and inhomogeneity [I. Maccari et al., 96, 060508(R) (2017)]. However, a full analysis of the data

including such non-trivial effects is not straightforward. Thus, we decided to discuss essentially the phase diagrams on the basis of the $T=0$ quantities. These general results can be understood without making reference to any specific theory, leaving for future work a detailed analysis of the temperature dependence.

Figure R10 : Temperature dependence of the normalized stiffness at $V_G=34V$ fitted using the theoretical expression in inset. The fit gives $\Delta(0)=23.4 \mu eV$.

Referee's comment : -Some less important points:

p.1 the authors mention the superconducting dome ending into a QCP – there are probably two QCP's (at both endpoints).

Indeed there is probably an other QCP on the overdoped side, however this latter is rarely reached and most of the superconducting domes are truncated.

p.5 ... with a 2π shift (Fig.2b) – not 3π .

We thank the referee for pointing out the mistake.

Changes : the reference to Fig 2b was corrected.

Reviewers' comments:

Reviewer #1 (Remarks to the Author):

I have read with interest the revised version of the paper "Competition between electron pairing and phase coherence in superconducting interfaces" by G. Singh et al.

I appreciate the Authors efforts to improve the manuscript in particular for what concern missing technical details concerning the experimental method. The results presented are certainly of general interest for the condensed matter community.

However, I have still few remarks which the Authors have to address before I can recommend publication on Nature Comm.

-Concerning the arguments given in the answer to my remarks on the experimental technique, I see that the Authors mention that their method cannot be used in the case of metallic superconductors, thus any additional test with known "reference" superconductor is unfeasible. I accept this argument. However, an independent test for the value of $J_s(0)$ (thus for L_1), can be performed by fitting the temperature dependence of the resonant frequency shift using the full BCS expression of the penetration depth in the dirty limit, or better analytical approximations which are valid in the full range [see for example Journal of Applied Physics 78, 1862 (1995)]. The fit will contain essentially only three parameters, $\lambda(0)$, $\Delta(0)$ and the ratio between the mean free path and the coherence length (ξ and eventually T_c). This will allow also to independently check not only the gap value but also the overall value of $J_s(0)$, which determine the scale of frequency shift from T_c to $T=0$.

Indeed, the fit (Fig. R8) performed by the Authors clearly is unable to reproduce the data, for at least two reasons. The expression used is not a good approximation above $T_c/2$ and the T_c to be used should be a value somewhat lower than the one used (In Fig. R8, also looking at the R vs T data, it seems that $T_c(R=0)$ is 0.13 K-14 K). I also notice that in the underdoped region of the phase diagram, if the 2DEG is actually a network of Josephson coupled SC islands, the effective penetration depth (thus the effective J_s) is clearly higher because one has to consider also the Josephson contribution.

In conclusion, in my opinion the Authors should make an additional effort to get a better fit of the data of the overdoped sample (at least), which the Authors claim to behave essentially as BCS superconductor.

I would like to finally remark, that the argument given to explain the differences between the fit performed and the data above $T_c/2$, in terms of possible jump of J_s due to a possible KT scenario is very doubtful, and I would suggest the Authors to be very cautious about that. It is certainly possible that a KT scenario applies to this 2D-superconductor, however the jump they refer to is about half $J_s(0)$!

-I have an additional remark, which unfortunately is quite crucial for what concern the possibility to generalize the results.

In my previous report, I was surprised about the vertical scale of Fig. 5d. Effectively, also looking at the details of the Hall effect curves, in agreement with the Authors response to my observation, it seems that the vertical scale of Fig.5d was correct. However, this poses a serious problem, since the carrier density of LAO/STO samples, with reduced number of oxygen vacancies, is at least a factor 5 or more lower [Nature 456, 624 (2008), Nano Lett. 16, 6130 (2016), Nature Communications 3, 1129 (2012).]. Here I see an inconsistency between the characteristics of the samples studied here and those of typical LAO/STO samples (reported by different groups). Thus the conclusions drawn cannot be generalized. This difference should be highlighted in the main text.

In support of their data, the Authors state that the carrier density is of the order of the value expected from the polar catastrophe scenario. This is highly debated, also in consideration of the

large discrepancy between the carrier density measured by many groups and the one expected.

Response to referee A

We thank the referee for his/her critical reading of our manuscript and suggestions of improvement. We detail below our response to his/her two remaining comments, together with the changes we made on the revised version of the article. We hope that this new version will be suitable for publication.

Comment 1: *-Concerning the arguments given in the answer to my remarks on the experimental technique, I see that the Authors mention that their method cannot be used in the case of metallic superconductors, thus any additional test with known "reference" superconductor is unfeasible. I accept this argument. However, an independent test for the value of $J_s(0)$ (thus for $L1$), can be performed by fitting the temperature dependence of the resonant frequency shift using the full BCS expression of the penetration depth in the dirty limit, or better analytical approximations which are valid in the full range [see for example Journal of Applied Physics 78, 1862 (1995)]. The fit will contain essentially only three parameters, $\lambda(0)$, $\Delta(0)$ and the ratio between the mean free path and the coherence length (and eventually T_c). This will allow also to independently check not only the gap value but also the overall value of $J_s(0)$, which determine the scale of frequency shift from T_c to $T=0$.*

Indeed, the fit (Fig. R8) performed by the Authors clearly is unable to reproduce the data, for at least two reasons. The expression used is not a good approximation above $T_c/2$ and the T_c to be used should be a value somewhat lower than the one used (In Fig. R8, also looking at the R vs T data, it seems that $T_c(R=0)$ is 0.13 K-14 K). I also notice that in the underdoped region of the phase diagram, if the 2DEG is actually a network of Josephson coupled SC islands, the effective penetration depth (thus the effective J_s) is clearly higher because one has to consider also the Josephson contribution.

In conclusion, in my opinion the Authors should make an additional effort to get a better fit of the data of the overdoped sample (at least), which the Authors claim to behave essentially as BCS superconductor.

I would like to finally remark, that the argument given to explain the differences between the fit performed and the data above $T_c/2$, in terms of possible jump of J_s due to a possible KT scenario is very doubtful, and I would suggest the Authors to be very cautious about that. It is certainly possible that a KT scenario applies to this 2D-superconductor, however the jump they refer to is about half $J_s(0)$!

Our response: In Fig. R8 of our previous report, we proposed to fit the temperature dependence of $J_s(T)$ using an approximate expression. Whereas a good agreement was obtained for the low temperature part of the curve, a strong deviation was observed for $T > T_c/2$ that we related to a BKT-like behavior. As suggested by the referee, we perform the same fitting procedure using a full BCS expression. The result, shown in Figure 1, confirms that the low temperature part of the curve is in agreement with BCS theory and gives a zero-temperature BCS gap $\Delta(0) \approx 24 \mu\text{eV}$ very close to the gap value extracted from the $J_s(0)$ and R_n in the article ($\approx 22 \mu\text{eV}$).

Extracting $J_s(0)$ (or $\lambda(0)$) directly from the temperature dependence of the resonance frequency $\omega_0(T)$ requires the knowledge of some physical parameters of the resonant circuit. This is valid for any type of resonant set-up. For instance, in the reference mentioned by the referee (Journal of Applied Physics 78, 1862 (1995)), there are three parameters that characterize the distributed ring resonator: diameter D , thickness d and dielectric constant of

the substrate ϵ_{eff} . The values of these parameters determine the total capacitance C and geometrical inductance L of the resonator and thus the resonance frequency ($\omega_0 = 1/\sqrt{LC}$). In this article, it seems that the authors fit $\omega_0(T)$ by considering that ϵ_{eff} is a fitting parameter and that the geometrical parameters (D, d) are known. This gives access to ϵ_{eff} and $\lambda(0)$. Our situation is quite equivalent, but instead of using a distributed resonator, we built a lumped-element resonator in which the inductance L_1 is a commercial SMD chip and the capacitance C_{sto} is the one associated with the STO substrate. As a result, we also need to know some physical parameters of the resonant circuit to perform a fit of $\omega_0(T)$, and to extract $J(0)$ (or equivalently $\lambda(0)$). Because it is a lumped-element circuit, we do not need to specify the geometry of the resonator and the dielectric constant (as in the article mentioned above) but instead we specify directly the value of L_1 . Note that, formally, L_1 is nothing else than a tiny wire whose inductance is also determined by the geometry, like in the ring resonator. In our case, the value of C_{sto} (or equivalently the dielectric constant) can be extracted from the fit of $\omega_0(T)$ but it is easier to deduce it directly from the normal state resonance ($1/\sqrt{L_1 C_{\text{sto}}}$). This is because in our set-up, the resonance frequency is also measurable in absence of superconductivity. $J_s(T)$ can therefore be directly related to $\omega_0(T)$ with a rather good accuracy (despite the overall difficulty of the experiment) as described in our previous response letter.

Figure 1 : BCS and BKT fit of the stiffness including finite-frequency effects and small inhomogeneity for $V_g=34\text{V}$ (see text for details).

Let us now discuss the high temperature part of the $J_s(T)$ curve. As we mentioned in the previous reply, the fit of the stiffness is not an easy task since the BKT behavior expected in this system is absolutely not conventional. To explain this in more detail we show in Fig. 1 the BKT fit of the stiffness based on what has been discussed so far in the literature within the context of conventional disordered thin films [R1,R2,R3,R4,R5]. First of all, it must be stressed that the BCS fit itself is *not* expected to reproduce the data in the presence of a BKT transition. Indeed, the BCS temperature T_{BCS} must be *higher* than the real T_c which is driven by BKT physics. The BCS fit J_s^{BCS} can only matches the experimental J_s^{exp} at low enough

temperatures, where vortex-antivortex pairs cannot be thermally excited. When bound pairs are created the stiffness J_s starts to deviate from the BCS calculation, even before the temperature where the real BKT jump due to vortex-pairs unbinding occurs. Then the deviations from the BCS fit occur in general at a temperature *lower* than the one where the BKT transition occurs. This effect is larger for smaller vortex fugacity, as it has been observed experimentally in disordered thin films [R1,R2,R3]. In the pure homogeneous case at zero frequency one can then obtain the red dashed line in Fig. 1. However, in real materials and in particular in interfaces the unavoidable inhomogeneity of the transition smears out the BKT jump. In addition, finite-frequency effects can lead to a finite stiffness even above the T_{BCS} expected for the homogeneous case.

How these two effects cooperate to modify the stiffness is not an easy problem to address. In Fig. 1 we show the results obtained following the procedure outlined in Ref [R4]. The stiffness is computed in the effective-medium-approximation using a Gaussian distribution of the local J_s^i values with variance $\sigma = 0.08\bar{J}_s$. As one can see, the trends are already the ones observed: a further enhancement of the downturn and a tailish behavior. To improve the quantitative agreement one should investigate wider distributions of local stiffnesses and the effects of spatial correlations for the disorder, not included in this calculation. Indeed, recent Monte Carlo simulations on the XY model (reported in Fig. 2 below) have shown that a "granular" superconducting landscape can induce a significant suppression of the stiffness much before than the universal line, due to the large proliferations of vortex-antivortex pairs in the low-stiffness regions. In these simulations the inhomogeneity of the local stiffness has been indeed inferred from the properties observed experimentally in strongly-disordered conventional superconductors, as e.g. NbN. An analogous analysis tailored to describe the inhomogeneity expected for LAO/STO interfaces is currently under investigation. This could shed also new light on the last point raised by the Referee, i.e. the interplay between inhomogeneities at different length scale (inside each SC puddle or between SC puddles)"

Figure 2 : Monte Carlo results for the stiffness of the XY model in the clean case (blue), in the presence of uncorrelated disorder (green) and correlated disorder (red). The spatial correlation significantly broadens the sharp jump found in the other two cases. Figure taken from Ref. [R5]

The above discussion demonstrates that the temperature dependence of the stiffness near T_c is a difficult problem that we are currently investigating and which goes well beyond the scope of the present manuscript. However, we stress once more that this issue is independent on the analysis of the energy scales at $T=0$ provided in our manuscript. The BKT physics, due to (classical) transverse phase fluctuations, is relevant near T_c , not at $T=0$. The BCS fit works perfectly at low temperatures and it provides a good estimate of the $T=0$ value of the excitation gap, which matches the BCS value.

Comment 2: *I have an additional remark, which unfortunately is quite crucial for what concern the possibility to generalize the results. In my previous report, I was surprised about the vertical scale of Fig. 5d. Effectively, also looking at the details of the Hall effect curves, in agreement with the Authors response to my observation, it seems that the vertical scale of Fig.5d was correct. However, this poses a serious problem, since the carrier density of LAO/STO samples, with reduced number of oxygen vacancies, is at least a factor 5 or more lower [Nature 456, 624 (2008), Nano Lett. 16, 6130 (2016), Nature Communications 3, 1129 (2012)]. Here I see an inconsistency between the characteristics of the samples studied here and those of typical LAO/STO samples (reported by different groups). Thus the conclusions drawn cannot be generalized. This difference should be highlighted in the main text. In support of their data, the Authors state that the carrier density is of the order of the value expected from the polar catastrophe scenario. This is highly debated, also in consideration of the large discrepancy between the carrier density measured by many groups and the one expected.*

Our response : The mechanism at the origin of the electronic reconstruction in LAO/STO interfaces is still a matter of debate in the community. The polar catastrophe scenario, which is often put forward, has the advantage to explain the 4 u.c. critical thickness for the 2-DEG creation that is observed experimentally. It predicts the transfer of 1/2 per u.c. at the interface corresponding to a carrier density of approximately $3.3 \cdot 10^{14}$ e/cm² (not too far from our). However, as correctly mentioned by the referee, this value is higher than what is measured experimentally, calling for caution. Indeed, conducting LAO/STO interfaces can be found in the literature with a broad range of carrier densities depending on the specific growth method of each group, typically from $1 \cdot 10^{13}$ to around $1 \cdot 10^{14}$ e/cm² (for example $1 \cdot 10^{14}$ at maximum doping in *Nano Lett. 16, 6130 (2016)*, $8.7 \cdot 10^{13}$ e/cm² at zero doping in PRB 88, 241301 (R) (2013), $8 \cdot 10^{13}$ e/cm² at max doping in Nat. Commun. 6, 6028 (2015)). An alternative mechanism, which combines the polar catastrophe and oxygen vacancy doping, was proposed by Yu and Zunger to reconcile the various experimental observations [Nature Communications, 5, 5118 (2014)]. However, there is not consensus on this issue, and, in absence of clearly identified electronic reconstruction mechanism, we do not know what should be the carrier density in a LAO/STO interface. For the type of LAO/STO sample that we used in this article, we typically obtain a carrier density at maximum doping close to $1-2 \cdot 10^{14}$ e/cm², which indeed corresponds to the upper limit of the doping range commonly observed. We have added this information in the text to avoid any misunderstanding. Nevertheless, our sample presents the same properties than other LAO/STO samples, including, in particular, a superconducting dome and a gate tunable Rashba spin-orbit coupling (see also PRB 96, 024509 (2017) for instance). We are therefore confident that the results presented in the manuscript are relevant for the community working on superconducting oxides interfaces, and more generally, for the understanding of low-

dimensional superconducting systems.

Changes : we have added the following sentence on the carrier density (page 9, last paragraph) : However, in our case n_s^{2D} is lower despite a much higher carrier density $n \approx 1.8 \cdot 10^{14} \text{ e}^- \cdot \text{cm}^{-2}$ at maximum doping) which corresponds to the upper limit of the doping range commonly observed in $\text{LaAlO}_3/\text{SrTiO}_3$ interfaces.

[R1] A. Kamlapure, M. Mondal, M. Chand, A. Mishra, J. Jesudasan, V. Bagwe, L. Benfatto, V. Tripathi, and P. Raychaudhuri, *Appl. Phys. Lett.* 96, 072509 (2010).

[R2] M. Mondal, S. Kumar, M. Chand, A. Kamlapure, G. Saraswat, G. Seibold, L. Benfatto, and P. Raychaudhuri, *Phys. Rev. Lett.* 107, 217003 (2011).

[R3] Jie Yong, T. Lemberger, L. Benfatto, K. Ilin, M. Siegel, *Phys. Rev. B* 87, 184505 (2013).

[R4] Rini Ganguly, Dipanjan Chaudhuri, Pratap Raychaudhuri, and Lara Benfatto, *Phys. Rev. B* 91, 054514 (2015).

[R5] I. Maccari, L. Benfatto and C. Castellani, *Phys. Rev B* 96, 060508(R) (2017)

REVIEWERS' COMMENTS:

Reviewer #1 (Remarks to the Author):

I have read the response letter to my comments and I appreciate the efforts of the Authors to fit the temperature dependence of the superfluid stiffness using the full BCS formula in the answer to my remarks. As far as I understood the Authors decided to cut the BCS fitting from the manuscript and from the suppl. Information. I do not know if this is a wise choice, since it would be useful for a reader to know that, in spite of the remarkable coincidence between the BCS gap value and the experimental result in the overdoped region of the phase diagram, still there are discrepancies between the $T > T_c/2$ data of the superfluid stiffness and the BCS theory. I think that a sentence summarizing this fact is important, since it shows that still a lot of work is needed to fully understand superconductivity in LAO/STO, even in the overdoped region of the phase diagram.

Response to referee A

We are delighted to read that the referee recommends publication. We detail below our response to his/her last remaining comment, together with the changes we made on the revised version of the article.

Comment : *I have read the response letter to my comments and I appreciate the efforts of the Authors to fit the temperature dependence of the superfluid stiffness using the full BCS formula in the answer to my remarks. As far as I understood the Authors decided to cut the BCS fitting from the manuscript and from the suppl. Information. I do not know if this is a wise choice, since it would be useful for a reader to know that, in spite of the remarkable coincidence between the BCS gap value and the experimental result in the overdoped region of the phase diagram, still there are discrepancies between the $T > T_c/2$ data of the superfluid stiffness and the BCS theory. I think that a sentence summarizing this fact is important, since it shows that still a lot of work is needed to fully understand superconductivity in LAO/STO, even in the overdoped region of the phase diagram.*

Changes :

- A sentence on the gap energy determination has been added in the "Superfluid stiffness and gap energy" section : *In the OD regime, we also checked that the gap value extracted from a BCS fit of the temperature dependence of J_s^{exp} matches Δ_s^{exp} obtained by Eq. (2) (Supplementary Note 3 and Supplementary Fig. 6).*
- A Supplementary Figure (6) showing the temperature dependence of J_s^{exp} fitted by the BCS model has been added in the Supplementary Material.
- A Supplementary Note (3) has been added to the Supplementary Material to discuss briefly the temperature dependence of J_s^{exp} in the context of BCS and BKT theories.